



# Nonlinear behavior of organic aerosol in biomass burning plumes: a microphysical model analysis

Igor B. Konovalov[1], Matthias Beekmann[2], Nikolai A. Golovushkin[1], and Meinrat O. Andreae[3,4,5]

[1]Institute of Applied Physics, Russian Academy of Sciences, Nizhniy Novgorod, Russia
[2]Laboratoire Interuniversitaire des Systèmes Atmosphériques (LISA), UMR7583, CNRS, Université Paris-Est-Créteil, Université de Paris, Institut Pierre Simon Laplace, Créteil, France
[3]Max Planck Institute for Chemistry, Mainz, Germany
[4]Scripps Institution of Oceanography, University of California San Diego, La Jolla, CA 92093, USA
[5]Department of Geology and Geophysics, King Saud University, Riyadh, Saudi Arabia

*Correspondence to*: Igor B. Konovalov (konov@appl.sci-nnov.ru)

**Abstract.** Organic aerosol (OA) is a major component of smoke plumes from open biomass burning (BB). Therefore, adequate representation of the atmospheric transformations of BB OA in chemistry-transport and climate models is an important prerequisite for accurate estimates of the impact of BB emissions on air quality and climate. However, field and laboratory studies of atmospheric transformations (aging) of BB OA have yielded a wide diversity of observed effects. This

diversity is still not sufficiently understood and thus not addressed in models. As OA evolution is governed by complex nonlinear processes, it is likely that at least a part of the observed variability of the BB OA aging effects is due to the factors associated with the intrinsic nonlinearity of the OA system. In this study, we performed a numerical analysis in order to gain a deeper understanding of such factors. We employ a microphysical dynamic model that represents gas-particle partitioning and OA oxidation chemistry within the volatility basis set (VBS) framework and includes a schematic parameterization of

BB OA dilution due to dispersion of an isolated smoke plume. Several VBS schemes of different complexity, which have been suggested in the literature to represent BB OA aging in regional and global chemistry-transport models, are applied to simulate BB OA evolution over a five-day period representative of a BB aerosol lifetime in the dry atmosphere. We consider the BB OA mass enhancement ratio (EnR), which is defined as the ratio of the mass concentration of BB OA to that of an inert tracer and allows us to eliminate the linear part of the dilution effects. We also analyze the behavior of the

hygroscopicity parameter, κ, that was simulated in a part of our numerical experiments. As a result, five qualitatively different regimes of OA evolution are identified, which comprise (1) a monotonic saturating increase of EnR, (2) an increase of EnR followed by a decrease, (3) an initial rapid decrease of EnR followed by a gradual increase, (4) an EnR increase between two intermittent stages of its decrease, or (5) a gradual decrease of EnR. We find that the EnR for BB aerosol aged from a few hours to a few tens of hours typically increases for larger initial sizes of the smoke plume (and therefore, smaller

dilution rates) or for lower initial OA concentrations (and thus more organic gases available to form secondary OA). However, these dependencies can be weakened or even reversed, depending on the BB OA age and on the ratio between the fragmentation and functionalization oxidation pathways. Nonlinear behavior of BB OA is also exhibited in the dependencies



of κ on the parameters of the plume. Application of the different VBS schemes results in large quantitative and qualitative differences between the simulations, although our analysis suggests also that the main qualitative features of OA evolution simulated with a complex two-dimensional VBS scheme can also be reproduced with a much simpler scheme. Overall, this study indicates that the BB aerosol evolution may strongly depend on parameters of the individual BB smoke plumes (such as the initial organic aerosol concentration and plume size) that are typically not resolved in chemistry transport models.

## 1 Introduction

Atmospheric aerosol is known to play an important role as a climate driver on global and regional scales and to adversely affect human health. A large fraction of the aerosol mass is typically represented by organic components forming liquid, amorphous, or glassy particulate matter, which here is referred to as organic aerosol (OA). As a climate forcer, OA scatters solar radiation and provides cloud condensation nuclei, thus directly and indirectly contributing to cooling of the atmosphere on the global scale (IPCC, 2013; Lelieveld et al., 2019), although part of it, so called brown carbon, can absorb sunlight, thus contributing to warming (see, e.g., Andreae, and Gelencsér, 2006; Feng et al., 2013; Jo et al., 2016). On a regional scale, of particular significance is the cooling effect of OA on climate in the Arctic (Sand et al., 2015), opposing the rapid increase of surface temperature that has been observed in recent decades (Bekryaev et al., 2010). As an agent of air pollution, OA constitutes a considerable fraction of fine particulates ($PM_{2.5}$) (Jimenez et al., 2009) that causes human health disorders and premature deaths (Pope et al., 2009; Burnett et al., 2018; Lelieveld et al., 2019). However, as evidenced by the large differences between the OA atmospheric budgets evaluated with different models and also by considerable discrepancies between simulations and observations of OA (see, e.g., Tsigaridis et al., 2014; Bessagnet et al., 2016; Tsigaridis and Kanakidou, 2018), the current knowledge of OA sources and its atmospheric transformations is still deficient, and corresponding modeling representations are very imperfect.

Open biomass burning (BB), i.e., vegetation fires and agricultural waste burning, provides a major source of OA on the global scale. Specifically, it has been estimated that BB emissions of primary OA (POA), which typically constitutes the predominant fraction of BB aerosol, contribute about 70 % of total POA emissions (Bond et al., 2013). In recent years, numerous studies have been aimed at investigating and modeling sources (e.g., May et al., 2013; Jathar et al., 2014; Konovalov et al., 2015; van der Werf et al., 2017), radiative effects (e.g., Saleh et al., 2013; 2015; Archer-Nicholls et al., 2016; Pokhrel et al., 2017; Yao et al., 2017), and atmospheric transformations (e.g., Cubison et al., 2011; Jolleys et al., 2012; Forrister et al., 2015; Shrivastava et al., 2015; Konovalov et al., 2015; 2017; Tsimpidi et al., 2018; Theodoritsi and Pandis, 2019 ) of BB OA and its components.

Both laboratory experiments and ambient observations suggest that the mass concentration of BB OA may undergo major, yet highly diverse, changes as a result of its aging under typical atmospheric conditions. These changes are commonly evaluated by means of the BB OA mass enhancement ratio (EnR), which is usually defined as the normalized ratio of the BB OA mass concentration to the concentration of an inert BB tracer. The normalization makes the EnR for freshly emitted





aerosol equal to unity. In particular, considerable increases (in many cases exceeding a factor of 2) of EnR were found in smog chamber experiments after a few hours of photochemical aging of smoke from wood or grass burning (e.g., Grieshop et al., 2009; Hennigan et al., 2011; Tiitta et al., 2016; Ciarelli et al., 2017a; Fang et al., 2017; Ahern et al., 2019), although there has been a large diversity between results of individual chamber experiments. As a result of aircraft experiments

conducted in North America around Mexico City and on the Yucatan Peninsula, significant increases in EnR have been reported by DeCarlo et al. (2008) and Yokelson et al. (2009) for aging BB plumes (from about 30 % up to a factor of 2). Konovalov et al. (2015) identified a substantial increase (by a factor of 2) in the enhancement ratio for mass concentration of particulate matter in smoke plumes after 1-2 days of transport over regions of Eastern Europe. A similar major increase in the enhancement ratio for BB aerosol mass concentration, but over about 15 hours, of photochemical oxidation of BB

plumes was deduced from an analysis of satellite measurements of aerosol optical depth (AOD) over Siberia (Konovalov et al., 2017). Based on several years of continuous measurements of BB OA in an African savannah, Vakkari et al. (2018) found that EnR more than doubles on average after three hours of daytime aging. However, there is also evidence that EnR may decrease or remain nearly constant in aging smoke plumes. For example, based on aircraft measurements, Akagi et al. (2012) identified a sharp decrease of EnR during the first hour after emissions. Using data from several field campaigns

conducted in Australia, North America, and western Africa, Jolleys et al. (2012; 2015) found that the BB OA enhancement ratios in highly aged BB plumes (typically transported between three and six days before the measurements were taken) were consistently smaller than those in the fresh plumes. The aforementioned analysis of satellite data (Konovalov et al., 2017) suggested evidence for a gradual decrease of EnR after its initial strong increase. At the same time, several observational studies (e.g., Capes et al., 2008; Brito et al., 2014; Sakamoto et al., 2015; May et al., 2015; Zhou et al., 2017) did not reveal

any significant net changes of EnR in aged BB plumes.

Numerous studies reported major changes in the chemical composition of BB OA due to its aging (e.g., DeCarlo et al., 2008; Cubison et al., 2011; Pratt et al., 2011; Jolleys et al., 2012; 2015; Brito et al., 2014; May et al., 2015; Bertrand et al., 2018; Lim et al., 2019) regardless of whether or not significant net changes were detected in the BB OA mass concentration. On the one hand, BB OA aging has been found to be typically associated with a rapid decay (over a period of a few hours under

typical atmospheric conditions) of some key chemical compounds contributing to POA (such as, e.g., levoglucosan): using aerosol mass spectrometry, such a decay can be inferred from a decrease of the mass fragment signatures at $m/z$ 60 (e.g., May et al., 2015) as well as from a more comprehensive analysis at the molecular level (Bertrand et al., 2018). On the other hand, atmospheric processing of BB OA has been reported to result in strong enhancements of the oxidation state of the organic matter. The increases of the O:C ratio (due to addition of, e.g., alcohol and carbonyl groups) are usually inferred

from increases in the mass spectrometric signal at $m/z$ 44 (e.g., Brito et al., 2014; May et al., 2015; Fang et al., 2017) and can be indicative of secondary organic aerosol (SOA) formation. Note that changes in the chemical composition and oxidation state of OA particles can affect their hygroscopic and optical properties (e.g., Lambe et al., 2011; Adler et al., 2011; Akagi et al., 2012; Fan et al., 2019) which need to be adequately specified in chemistry-transport and climate models.



Useful insights into the possible reasons behind the large variability of the EnR values reported earlier for aging BB aerosol have been provided by recent smog chamber experiments (Ahern et al., 2019; Lim et al., 2019) that revealed a strong dependence of SOA formation on variable initial concentrations of organic gases. These experiments, however, do not rule out the possibility that there are some other factors contributing significantly to the observed diversity of changes in EnR

during the atmospheric lifetime of BB aerosol. In view of significant nonlinear interactions of the processes affecting properties, formation and evolution of SOA (Shrivastava et al., 2017), it seems reasonable to expect that the diversity of observational findings concerning BB aerosol atmospheric aging can in part be due to nonlinear behavior of OA transformations. By the nonlinear behavior, we mean here any manifestations of a dependence of the relative rate of change of OA mass concentration at a given moment of time on the mass concentration of OA itself at this or previous moments.

In this study, we investigate qualitative nonlinear features of the behavior of OA within an isolated BB plume and attempt to reconcile some of the diverse observational findings concerning BB aerosol aging effects from a theoretical viewpoint. To this end, using some routines and interfaces of the CHIMERE chemistry transport model (Menut et al., 2013), we developed and employed a microphysical dynamic (box) model of organic aerosol (MDMOA). While three-dimensional chemistry transport models are intended to provide the best possible quantitative representation of the evolution of OA and its gaseous

precursors from various anthropogenic and natural sources, the principal purpose of MDMOA is to isolate and simulate, under fixed ambient conditions, the effects of key processes responsible for chemical and physical transformations of OA from other complex processes affecting evolution of OA in the real atmosphere (such as mixing with aerosols and their gaseous precursors from multiple sources, vertical advection, dry and wet deposition, in-cloud processing, etc.). In this sense, our study is similar to several previous studies employing box models to study OA processes (e.g., Camredon et al.,

2007; Lee-Taylor et al., 2011; 2015; Lannuque et al., 2018). Note that compared to smog chamber and dedicated field studies, a box model analysis enables a much more comprehensive examination of the parameter space of the BB OA system.

It has been proposed that complex atmospheric transformations of OA (regardless of its origin), including SOA formation, can be adequately represented in chemistry transport models within the volatility basis set (VBS) modeling framework

(Donahue et al., 2006; 2011; 2012a; Robinson et al., 2007). This framework has been implemented in MDMOA. The VBS method involves splitting semi-volatile organic compounds (SVOCs) and the more volatile intermediate-volatility organic compounds (IVOCS), into several classes with respect to volatility and applying the absorptive partitioning theory (Pankow et al., 1994) to distribute the organic compounds between gas phase and particles. The SVOCs and IVOCs can also be distributed between several model types, depending, e.g., on their oxidation state (O:C ratio), origin (e.g., primary or

secondary, anthropogenic or biogenic, etc.), and photochemical age (Donahue et al., 2012a,b; Shrivastava et al., 2013; Tsimpidi et al., 2018). Representing the processes involving SVOCs and IVOCs within the VBS framework has been shown to allow improving the performance of simulations of OA from vegetation fires with respect to simulations using the "conventional" OA modeling framework, in which these processes are basically disregarded and only specific volatile


organic compounds (VOCs) are considered as precursors of SOA (Hodzic et al., 2010; Shrivastava et al., 2015; Konovalov et al., 2015; 2017). Based on simulations using the VBS method, it has also been argued (Konovalov et al., 2015; 2017) that disregard for the BB OA aging processes might be one of the main reasons for a strong underestimation of aerosol optical depth in BB plumes by chemistry transport models using the conventional representation of OA evolution (e.g., Tosca et al.,

2013; Konovalov et al., 2014; 2018; Reddington et al., 2016; Petrenko et al., 2017). It should be noted, however, that the representation of the BB OA evolution within the VBS framework in chemistry transport models is still associated with major uncertainties: while a variety of VBS schemes of different complexities have been suggested for BB OA modeling (e.g., Grieshop et al., 2009; Koo et al., 2014; Shrivastava et al., 2015; Ciarelli et al., 2017a; Tsimpidi et al., 2018), any of these schemes have only partially been constrained by laboratory or ambient measurements. In view of these uncertainties,

we performed our analysis using several different available VBS schemes.

Based on simulations of the first five hours of BB OA evolution with a similar microphysical box model, Bian et al. (2017) showed that apart from oxidation, evaporation and condensation of SVOCs, BB OA dynamics is strongly affected by the dilution process. Accordingly, dilution is also taken into account in our model. Following Bian et al. (2017), we approximate the dilution rate as a function of the initial plume size by using the formulations of the stationary Gaussian dispersion model

and analyze the dependence of the BB OA mass enhancement ratio on the initial plume size, which controls the dilution rate. However, we considerably extend the period of analysis (up to five days), and instead of the simple single-step oxidation scheme used by Bian et al. (2017) to analyze short-run smoke chamber experiments and to study the atmospheric implications of their results, we use several multi-step oxidation schemes that have been suggested for modeling of BB OA specifically within chemistry transport models. Along with the dynamics of EnR, we consider the evolution of the

hygroscopicity parameter, κ, which is commonly used to characterize the uptake of water by aerosol particles and their cloud condensation nucleus (CCN) activity (e.g., Petters and Kreidenweis, 2007; Chang et al., 2010; Mikhailov et al., 2015). The evolution of the hygroscopicity parameter, however, is not the main focus of this study.

By using MDMOA, we address the following questions. What are manifestations of the OA system's nonlinearity in the dependencies of EnR and κ on the initial size and initial density of a smoke plume? Can variability of the parameters of the

plume lead to qualitatively different types of BB OA evolution? Can differences between available VBS schemes be associated with qualitatively different responses of EnR to variations in the plume's parameters? It should be emphasized that our simple model and its application in this study are not intended to reproduce any realistic scenarios of atmospheric evolution of BB OA in a quantitatively accurate way. Instead, we focus our analysis on identification of possible qualitative features of the BB OA behavior, which may have a sufficiently general character. We believe that the results of this kind of

analysis can be useful as theoretical guidance for corresponding experimental studies and for improving parameterizations of BB OA processes in chemistry transport models.



## 2 Model and method description

### 2.1 Microphysical dynamic model of organic aerosol (MDMOA): dynamic equations

The CHIMERE based box mode MDMOA is intended to represent the following processes: (1) growth and evaporation of multi-disperse particles of OA due to partitioning of SVOCs between gas phase and particles, (2) gas-phase oxidation of

VOCs, IVOCs and SVOCs, and (3) atmospheric dilution of OA. The model also includes a representation of coagulation, but this process has not been taken into account in the present study. MDMOA has been developed by adopting and modifying several modules of the CHIMERE chemistry transport model (Menut et al., 2013), including the routines implementing the Gauss-Seidel iteration scheme (Verwer et al., 1994) to solve a set of dynamic equations, a sectional representation of the OA mass absorption and evaporation processes (Gelbard and Seinfeld, 1980), and some model interfaces facilitating

modifications of a simulation configuration as well as providing simulation outputs in a convenient NetCDF format. Dynamic mass transfer equations for a semi-volatile species $s$ in a particle size section $l$ are formulated as follows:

$$\frac{dC_s^l}{dt} = \frac{2}{3}\pi d_p^l c\lambda F N_p^l\left(C_s^g - \mathrm{K}C_s^{eq}\right) + I_{l-1}^l + I_l^{l+1},\tag{1}$$

where $C_s^l$ is the condensed-phase mass concentration, $d_p^l$ is the particle diameter, $c$ and $\lambda$ are the mean velocity and free path of the organic molecule in the air, $F$ is the Fuchs-Sutugin correction factor, $N_p^l$ is the number of particles in the size bin $l$, $C_s^g$

is the instantaneous gas-phase concentration, K is the Kelvin effect factor, $C_s^{eq}$ is the equilibrium gas-phase concentration, $I_{l-1}^l$ and $I_l^{l+1}$ are the intersectional fluxes between the bins $l$-1 and $l$ and between the bins $l$ and $l$+1.

The molecular mean free path, the Fuchs-Sutugin correction factor and the Kelvin effect factor are evaluated using the conventional formulations (Seinfeld and Pandis, 2016):

$$\lambda = 3Dc^{-1},\tag{2}$$

$$F = \frac{1+Kn}{1+0.3773Kn+1.33Kn\left(\frac{1+Kn}{\alpha}\right)},\tag{3}$$

$$\mathrm{K} = \exp\left(\frac{4\sigma_p MW_s}{RT\rho_p d_p^l}\right),\tag{4}$$

where $D$ is the molecular diffusion coefficient, $Kn$ is the Knudsen number ($Kn$=$2\lambda/d_p^l$), $\alpha$ is the mass accommodation coefficient (that is assumed to be unity in all our simulations), $\sigma_p$ and $\rho_p$ are the surface tension and density of the particle material, $MW_s$ is the molecular weight, $R$ is the ideal gas constant and $T$ is temperature. Following the basic formulations for

the VBS framework, the gas-phase equilibrium concentration is expressed through the total mass concentration of SVOCs, $C^{tot}$, the mass fraction of a given species, $f_s$, the total mass concentration of OA particles, $C_{OA}$, and the saturation concentration $C_s^*$:

$$C_s^{eq} = C_s^* \frac{f_s C^{tot}}{C_{OA}}\left(1 + \frac{C_s^*}{C_{OA}}\right)^{-1}.\tag{5}$$



Note that the formulation of the mass flux term for transfer of SVOCs from and into the gas phase in Eq. (1) is essentially the same as that in the kinetic model used by May et al. (2013) to derive the volatility distributions for BB POA. Following May et al. (2013), we also assumed, for definiteness, that the diffusion coefficient, surface tension and the particle bulk density are equal to $5 \times 10^{-6}$ m$^2$ s$^{-1}$, 0.05 N m$^{-1}$ and $1.2 \times 10^3$ kg m$^{-3}$, respectively.

The intersectional fluxes are calculated according to Gelbard and Seinfeld (1980) as a combination of the weighed mass fluxes between the gas and particle phases for the bins $l$-1, $l$, and $l$+1. A concrete representation of the intersectional fluxes is not of significance in this study since they cannot, by definition, contribute to the mass balance (their sum over the all particle size bins equals zero), and we do not consider here the evolution of the particle size distribution. Furthermore, as argued below, the equilibration time scales determined by Eq. (1) are typically much smaller than the time scales associated

with oxidation of SVOCs and so our simulations are not sensitive to the particle size distribution.

The dynamics of the total concentration (both in the gas phase and in particles), $C_s^{tot}$, of a given SVOCs species is driven by the following mass balance equations:

$$\frac{dC_s^{tot}}{dt} = -\frac{1}{V_P}\frac{dV_P}{dt} C_s^{tot} - k_{OH}^s [OH]C_s^g + P_s, \qquad (6)$$

where $k_{OH}^s$ is the oxidation reaction rate, [OH] is concentration of hydroxyl radical, $V_P$ is the volume of a BB plume, and $P_s$
is the chemical production rate of $s$. The reaction rates and chemical processes specified in the model are described in the next section (Sect. 2.2). Note that Eqs. (6) for several different species compose an essentially nonlinear system. In particular, not only does $C_s^g$ in thermodynamic equilibrium depends nonlinearly on the total aerosol concentration, $C_{OA}$, in accordance with Eq. (5), but $C_{OA}$ itself also depends in a complex nonlinear manner on the total concentrations, $C_s^{tot}$, of all SVOCs. Furthermore, $P_s$ is determined by the gas phase concentrations of SVOCs, too, and therefore depends nonlinearly on
both $C_s^{tot}$ and $C_{OA}$.

Representation of the dilution process (described in Eq. (6) by the term proportional to $\frac{dV_P}{dt}$) in our simulations largely follows Bian et al. (2017). Specifically, we assume that all the species considered are uniformly distributed within a box with a half-width of $2\sigma_y$ across the wind direction and a half-thickness of $2\sigma_z$ in the vertical. The thickness of the box in the wind direction does not need to be explicitly specified in our simulations (but just for definiteness, it can be assumed to be equal to
1 m). The evolution of $\sigma_y$ and $\sigma_z$ is represented by the power law expressions according to Klug et al. (1969) (see also Seinfeld and Pandis, 2016) for the neutral (D) Pasquill atmospheric stability class. The plume is assumed to be transported along the wind direction with a constant speed of 5 m s$^{-1}$. The initial width of the plume ($4\sigma_y$) is considered as a control parameter, $S_p$, in our simulations. The initial plume width, $S_p$, can also be interpreted as the across-wind width of the area affected by the fire. The initial value of $\sigma_z$ is expressed as a function of $\sigma_y$. It is assumed that the plume's thickness in the
vertical direction ($4\sigma_z$) cannot exceed the mixed layer height, which is fixed at 2500 m: that is, once $\sigma_z$ calculated according to Klug et al. (1969) reaches 625 m, the plume is allowed to disperse only in the horizontal direction. Such a simple representation of the plume's evolution is by no means intended to be quantitatively accurate under any real conditions but is



used mainly to roughly characterize a dependence of the temporal scale of the dilution process on the horizontal spatial scale of a BB plume, especially during the first few hours of evolution.

## 2.2 Representations of BB OA oxidation processes and gas-particle partitioning in MDMOA

To take into account the existing ambiguity associated with the representation of the oxidation of organic matter within the VBS framework, we performed our simulations using several VBS schemes of varying complexity. A summary of the main features of the schemes and their reference codes is provided in Table 1. We also used a "conventional" OA scheme that assumes that POA is composed of non-volatile species. Below we describe the schemes in more detail.

The scheme "C17" has been described and evaluated by Ciarelli et al. (2017a, b). It is a relatively simple scheme which has been referred to as a hybrid 1.5-dimensional (1.5D-VBS) and is based on a similar scheme proposed by Koo et al. (2014). The idea behind this scheme is to characterize sources, volatilities, chemical transformation and oxidation state of a complex mixture of SVOCs by considering several surrogate species which are given average molecular composition and molecular weights. The scheme distinguishes between three sets of the surrogate species, i.e., the POA set (set 1), the set containing oxidation products from reactions of hydroxyl radical with semi-volatile gases from POA (set 2), and the SOA set (set 3) representing products of reactions of OH with any VOCs (and, implicitly, also IVOCs). Semi-volatile gases from sets 2 and 3 are also allowed to react with OH, with the volatility of the product being an order of magnitude lower than that of the reactant. The different reaction products have different molecular weights and are assumed to represent the net effects of the actual functionalization and fragmentation reactions (neither of which are specified explicitly). All SVOCs are distributed among 5 bins covering the volatility range from $10^{-1}$ to $10^{3}$ $\mu$g m$^{-3}$. The reaction rate ($k_{OH}$) is fixed at $4.0 \times 10^{-11}$ cm$^{3}$ molec$^{-1}$ s$^{-1}$ for all the oxidation reactions. Some parameters of the scheme have been optimized by fitting box-model simulations to the data from several aging experiments with BB aerosol from stove wood combustion (Ciarelli et al., 2017a). Based on the optimization results, the average ratio of the initial total mass concentrations of VOCs and SVOCs was set at 4.75. The scheme has been then implemented into a chemistry transport model and successfully evaluated against ambient measurements performed with an aerosol mass spectrometer (AMS) across Europe, specifically in situations where a considerable part of OA originated from residential wood burning (Ciarelli et al., 2017b). Note, however, that the composition and conditions of atmospheric aging of OA from residential wood burning are not necessary representative of those of OA from vegetation fires.

The scheme "K15" is a 1-dimensional (1D) scheme that has been introduced by Konovalov et al. (2015) in an air pollution case study to represent atmospheric aging of BB OA from the 2010 Russian fires. It combines one of the simplest 1D schemes (Grieshop et al., 2009), in which only functionalization reactions have been taken into account, and a more complex scheme in which both functionalization and fragmentation processes are taken into account (albeit in a very simplified manner) and which is referred below as the scheme "S15" (Shrivastava et al., 2013; 2015). The oxidation processes are described using a volatility grid that includes 7 bins ($10^{-2} \leq$ C* $\leq 10^{4}$ $\mu$g m$^{-3}$). The scheme distinguishes between oxidation of primary organic gases (POG), which is assumed to result only in functionalization, and oxidation of secondary organic gases





(SOG), which is assumed to include both functionalization and fragmentation branches. The products of the functionalization branch get their mass increased by 40 % and the volatility reduced by two orders magnitude with respect to those of the reactants. Specifically, oxidation of POG and SOG from volatility bin $i$ is represented as follows:

$$POG_{i>2} + OH \rightarrow 1.4\ SOG_{i-2}, \tag{7}$$

$$SOG_{i>2} + OH \rightarrow 0.5 \times 1.4\ SOG_{i-2} + 0.4\ SOG_{i=7} + 0.1\ LCN, \tag{8}$$

where LCN denotes the highly volatile low-carbon-number species that are the products of the fragmentation branch, and all the species are assumed to have the same molecular weight (250 g mole$^{-1}$). Along with LCN, the fragmentation branch yields SOG in the highest volatility bin. While LCN species are not allowed to participate in any reactions, SOG species can be reprocessed according to Eq. (8). Note that oxidation of SOGs results in a net increase of the organic mass, although Eq. (8)

formally corresponds to a fragmentation branching ratio (Jimenez et al., 2009) of 0.5. Note also that the simulations reported by Konovalov et al. (2015) included the transformation of condensed-phase SOA into non-volatile SOA (NVSOA) and indicated that this process had only a small impact on the simulated evolution of BB aerosol in the case considered. In view of a lack of robust knowledge about the condensed-phase processes (see also Section 4) and for consistency with the other numerical experiments performed in the present study, the transformation of SOA into NVSOA has been disregarded in our

simulations.

The scheme "S15" is a slightly modified version of the VBS scheme that was proposed by Shrivastava et al. (2013) and adopted, as a part of a global chemistry transport model, for BB aerosol modeling in a subsequent study (Shrivastava et al., 2015). Unlike the original VBS scheme by Shrivastava et al. (2013, 2015), where only five volatility classes are used for computational reasons, the volatility basis set in the S15 scheme is specified using the same seven volatility bins as in the

K15 scheme for the sake of easier interpretation of differences between the respective simulation results. The S15 scheme can be regarded as a quasi-2-dimensional scheme as it realizes a computationally efficient way to account for the increasing probability of fragmentation reactions with BB OA aging (and, implicitly, with increasing oxidation state) of BB OA by distinguishing between different generations, $n$, of SOA precursors. Specifically, while POGs and the first two generations of SOGs are assumed to undergo only functionalization reactions:

$$POG_{i>1} + OH \rightarrow 1.15\ SOG_{i-1,\ n=1}, \tag{9}$$

$$SOG_{i>1,\ n\leq 2} + OH \rightarrow 1.15\ SOG_{i-1,\ n+1}, \tag{10}$$

the third and further generations undergo both functionalization and fragmentation reactions:

$$SOG_{i,\ n\geq 3} + OH \rightarrow (1-\beta_{fr}) \times 1.15 \times SOG_{i-1,\ n+1} + \beta_{fr} \times (0.88 \times SOG_{i=7,\ n+1} + 0.12 \times LCN), \tag{11}$$

where $\beta_{fr}$ is the fragmentation branching ratio which is assumed to be equal, in this case, to 0.85.

According to Eqs. (9)-(11), the functionalization reactions of both POGs and SOGs yield SOG species in the next lower volatility bin and result in an increase of the molecular weight by 15 %. Similar to the K15 scheme, all the VBS species are assumed to have the same molar mass of 250 g mol$^{-1}$. Note that using the OA oxidation scheme described above, Shrivastava



et al. (2015) defined the two modeling configurations, FragSVSOA and FragNVSOA, with SOA treated as semi-volatile or non-volatile, respectively. The simulations performed with the S15 scheme in this study correspond only to the FragSVSOA configuration (that is, the NVSOA formation has not been included in MDMOA).

Scheme "T18" is a 2-dimensional (2D) VBS scheme that is adopted (with minor modifications) from Tsimpidi et al. (2018),

where it has been introduced as a part of the ORACLE v2.0 aerosol module of the ECHAM/MESSy Atmospheric Chemistry (EMAC) global model. The scheme represents the oxidation of BB OA on the 2D VBS grid constructed in the space of the volatility and the oxygen content (O:C ratio). The volatility dimension is discretized into 4 bins: $C^* = \{10^{-2}; 10^{0}; 10^{2}; 10^{4}\}$ µg m$^{-3}$ and the oxygen content dimension is divided into 11 O:C bins covering the O:C values from 0.2 to 1.2 with a constant step of 0.1. The smallest value of the O:C ratio is assumed to be representative of fresh BB emissions and is attributed to

POA. Each species is identified with a representative number of carbon atoms per molecule, $n_c$, and with a molecular weight, $MW$, evaluated using structure activity relationships (Pankow and Asher, 2008; Donahue et al., 2011) and an approximation of the hydrogen to carbon atomic ratio (Heald et al., 2010) as follows:

$$n_c = \frac{11.875 - \log_{10} C^*}{0.475 + 2.3(O:C) - 0.6(O:C)\left(1 + (O:C)\right)^{-1}} , \tag{12}$$

$$MW = (15(O:C) + 14)n_c \tag{13}$$

Oxidation of POG (or SOG) species is assumed to result in addition of two or three oxygen atoms to their molecules with an equal probability. The O:C ratio of the products is therefore evaluated as follows:

$$(O:C)_{product} = (O:C)_{reactant} + \frac{\Delta O}{(n_c)_{reactant}} , \tag{14}$$

where $\Delta O$ (equal to 2 or 3) is an assumed increment of the oxygen atomic content. The product is assumed to belong to the next lower volatility class with respect to the volatility class of the reactant. Similar oxidation reactions also apply to POG

and SOG species from the lowest volatility class, except that the products of these reactions keep the volatility of the reactants. Fragmentation reactions are not explicitly taken into account. As noted by Tsimpidi et al. (2018), neglecting fragmentation may result in overestimation of OA concentration at long aging timescales. However, it should also be noted that since $n_c$ can decrease as a result of an oxidation step in accordance with Eq. (12), the fragmentation pathway is, to some extent, taken into account in the T18 scheme implicitly.

Along with similar 2D-VBS schemes for anthropogenic and biogenic OA, the T18 scheme described above has been used for multi-year simulations of OA with the EMAC model, and the simulation results were compared against AMS measurements at urban downwind and rural environments in the Northern Hemisphere (Tsimpidi et al., 2018). However, the comparison results did not provide enough information about the performance of the T18 scheme in simulations of OA specifically from BB sources.

The last VBS scheme that we used, "T18f", is our modification of the original T18 scheme. It has been obtained in this study by adding explicit fragmentation pathways to the original T18 scheme, so that any POG or SOG species considered in the





T18 scheme is assumed to participate in both functionalization and fragmentation reactions. The reactions originally included in the T18 scheme are assumed (for simplicity) to represent only the functionalization pathways. The probability of a given pathway is controlled by the fragmentation branching ratio, $\beta_{frag}$, which is parameterized as follows (Jimenez et al., 2009; Donahue at al., 2012b):

$$\beta_{frag} = (O:C)^{1/4}. \tag{15}$$

Following Murphy et al. (2012), we assume that splitting of an organic molecule as a result of fragmentation reactions occurs with a uniform probability at any site throughout its carbon backbone. Accordingly, a fragmentation reaction of any species containing (according to Eq. 12) $n_c$ carbon atoms can potentially yield $(n_c-1)/2+1$ (if $n_c$ is an odd number) or $n_c/2$ (if $n_c$ is an even number) different (with respect to the atomic carbon content) products. Furthermore, consistently with the assumptions underlying the original T18 scheme, we assume that as a result of any oxidation reaction, one of the two fragments receives two or three additional oxygen atoms, and so its O:C ratio increases in accordance with Eq. (14), except that the atomic carbon number of the reactant in the right-hand part of the equation should be substituted for that of the product. The O:C ratio of the other fragment is kept the same as that of the reactant. If the calculated O:C ratio of a product exceeds 1.2 (that is, the maximum value covered by the O:C grid considered), this product is assumed to be irreversibly lost into the gas phase. All possible pairs of fragmentation products described above are introduced in the corresponding mass balance equations (see Eq. 6). The stoichiometric coefficients for the products in the functionalization, $k_s^{fn}$, and fragmentation, $k_s^{fr}$, pathways are evaluated as follows:

$$k_s^{fn} = (1 - \beta_{frag})/N_{ps}^{fn}, \tag{16}$$

$$k_s^{fr} = \beta_{frag}/N_{ps}^{fr}, \tag{17}$$

where $N_{ps}^{fn}$ and $N_{ps}^{fr}$ are the total numbers of possible products in the functionalization and fragmentation pathways, respectively.

Finally, we also consider a linear analogue to the above nonlinear representations of BB OA evolution. The scheme "LIN" is based on a simple oxidation scheme (Pun et al., 2006; Bessagnet et al., 2008) designed in the framework of the conventional approach to representation of OA evolution and SOA formation and implemented in the CHIMERE model. The key assumptions underlying this scheme are that POA is composed of nonvolatile species and that SOA is formed from oxidation of several specific ("traditional") volatile precursors that were identified earlier in smog chamber experiments (Odum et al., 1997). In this study, the original scheme described in detail by Menut et al. (2013) was simplified (linearized) by assuming that all oxidation products are nonvolatile. The emission factors from Andreae (2019) for the boreal forest were used to specify initial concentrations of POA precursors as a function of the initial concentration of BB OA. Note that the original SOA formation scheme from the CHIMERE model was earlier used in the simulations of evolution of BB aerosol from Russian fires (Konovalov et al., 2015; 2017; 2018) and was found to produce rather negligible amounts of SOA in BB plumes.





**Table 1: Reference codes and main features of the BB OA modeling schemes used in the simulations. SVOC: semi-volatile organic compound; VOC: volatile organic compound; POG: primary organic gas; SOG: secondary organic gas; POA: primary organic aerosol; SOA: secondary organic aerosol; $C^*$: saturation mass concentration; $\beta_{frag}$: fragmentation branching ratio.**

| Oxidation scheme | Key features | References |
|---|---|---|
| C17 (a hybrid 1.5D-VBS scheme) | Three sets of SVOCs to model oxidation of organics; five volatility classes ($0.1\,\mu g\,m^{-3} \leq C^* \leq 10^3\,\mu g\,m^{-3}$) to model gas-particle partitioning; an implicit representation of functionalization and fragmentation reaction pathways; one-bin shift in volatility of a product of oxidation reactions of POG or SOG with OH ($k_{OH}=4\cdot10^{-11}\,cm^3\,s^{-1}$) with respect to those of a reactant | Ciarelli et al. (2017a,b) |
| K15 (a 1D-VBS scheme) | An explicit representation of the functionalization and fragmentation branches ($\beta_{frag} = 0.5$) of each reaction of POG or SOG with OH ($k_{OH} = 2\cdot10^{-11}\,cm^3\,s^{-1}$); a two-bin shift in volatility and 40 % increase of the molecular weight for a product of the functionalization reaction pathway; seven volatility classes ($0.01\,\mu g\,m^{-3} \leq C^* \leq 10^4\,\mu g\,m^{-3}$) for both primary and secondary SVOCs | Konovalov et al. (2015) |
| S15 (a quasi 2D-VBS scheme) | An explicit representation of the functionalization and fragmentation branches with distinction between reactions involving POG and "fresh" SOG ($\beta_{frag} = 0$) and reactions involving "aged" SOG ($\beta_{frag} = 0.85$); a one-bin shift in volatility and 15 % increase of the molecular weight for a product of the functionalization pathway of each reaction of POG or SOG with OH ($k_{OH} = 4\cdot10^{-11}\,cm^3\,s^{-1}$); seven[1] volatility classes ($10^{-2}\,\mu g\,m^{-3} \leq C^* \leq 10^4\,\mu g\,m^{-3}$) for both primary and secondary SVOCs | Shrivastava et al. (2013; 2015) |
| T18 (a 2D-VBS scheme without fragmentation) | A representation of SVOCs on a 2D grid space covering four volatility classes ($C^*=\{10^{-2};\ 10^0;\ 10^2;\ 10^4\}\,\mu g\,m^{-3}$) and 11 linearly spaced oxygen content bins (O:C=$\{0.2, 0.3, \dots 1.2\}$); the SVOC molecular mass defined as function of $C^*$ and O:C; no explicit representation of fragmentation; a one-bin shift in volatility for a product of any oxidation reaction ($k_{OH} = 2\cdot10^{-11}\,cm^3\,s^{-1}$) | Tsimpidi et al. (2018) |
| T18f (a modified T18 scheme with fragmentation) | The same as the T18 scheme but with fragmentation reactions ($\beta_{frag} = $ (O:C)$^{1/4}$, a uniform probability of fragmentation across the backbone of an organic molecule) | Tsimpidi et al. (2018); Donahue at al. (2012b); Murphy et al. (2012) |
| LIN (a "linear" OA scheme) | POA and SOA composed of nonvolatile species; SOA formation from oxidation of several specific VOCs | Pun et al. (2006); Bessagnet et al. (2008); Menut et al. (2013) |

[1] Note that the original VBS scheme (Shrivastava et al., 2015) involves only five volatility classes

5   It should be stressed that the different VBS schemes outlined above certainly do not comprise all known mechanisms and pathways of oxidation of organic compounds composing BB OA. Nonetheless, consideration of even a limited spectrum of available representations of the BB OA aging processes allows us to get a useful insight into the uncertainty associated with simulations of BB OA aerosol evolution using the VBS framework. Some processes which could further enhance the diversity of our simulations of BB OA evolution are briefly discussed in Sect. 4.



### 2.3 Configuration of the numerical experiments and processing of output data

MDMOA was run using each VBS scheme described above for a period of 120 hours. This period is representative of the typical lifetime of BB aerosol in Siberia under conditions without precipitation (Paris et al., 2009). A part of the simulation period (7 hours per day) was assumed to correspond to nighttime conditions when any oxidation processes were disabled
(OH concentration in Eq. (6) was set to be zero); such a nighttime duration is typical, for instance, for central Siberia in summer.

The initial conditions for SVOCs in particles and in the gas phase correspond to the gas-particle equilibrium determined in accordance with the partitioning theory. Specifically, the gas-phase initial concentration for a species $s$ was calculated using Eq. (5) where the OA mass concentration was assumed to include, along with the BB fraction, background OA
concentration, $C_{bg}$, of 5 μg m$^{-3}$. The total mass concentration of SVOCs, $C^{tot}$, that is involved in Eq. (5) was evaluated as follows:

$$C^{tot} = \left. C_0 \middle/ \sum_i \frac{f_i}{\left(1+\frac{c_i^*}{c_0+c_{bg}}\right)} \right. , \tag{18}$$

where $f_i$ defines the mass fraction of all species in the bin $i$ of the volatility distribution, and $C_0$ is the initial BB OA concentration. $C_0$ is considered – along with the initial plume size, $S_p$ – as a control parameter in our simulations. Test
experiments (see Sect. 3.1) have shown that the characteristic time scales for the adjustment to the "local" thermodynamic equilibrium are short (seconds or minutes) compared to the time scales associated with the oxidation processes (hours). The background aerosol concentration was not affected by any process except for the intersectional fluxes and so was basically kept constant in all the simulations.

The volatility distribution for all our experiments with the K15 and S15 schemes (in which the volatility grid includes seven
bins) was adopted from the study by Konovalov et al. (2015): $f$ = {0.1; 0; 0.05; 0.05; 0.2; 0.15; 0.45} at a temperature of 298 K. This distribution is consistent (within the range of uncertainties) with the data from thermodenuder measurements of BB emissions (May et al., 2013). The volatility distributions for the C17, T18 and T18f schemes were obtained from the same distribution by disregarding or aggregating the corresponding volatility bins.

In the experiments with the C17, K15 and S15 schemes, the aerosol size distribution was modeled using 9 size bins covering
the range from 20 nm to 10 μm. To limit the computational time, the experiments with the T18 and T18f schemes were conducted using only 3 size bins. In all cases, we used a lognormal distribution with a mass mean diameter of 0.3 μm and a geometric standard deviation of 1.6 (Reid et al., 2005). While the size distribution can affect the time scales for evaporation and growth of particles, these time scales, as noted above, are much smaller than those associated with chemical aging. Accordingly, in all our simulations, the results of the simulations have been practically independent of the assumed particle
size distribution (except for a very minor influence of the Kelvin effect). Note that the representation of non-equilibrium





processes in accordance with Eq. (1) has been included in MDMOA mainly to enable simulations of evolution of BB aerosol optical properties in prospective studies.

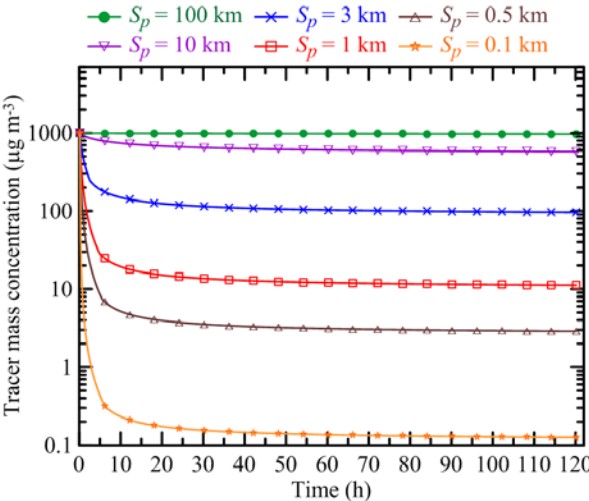

**Figure 1: Simulated evolution of the mass concentration of an inert tracer for different values of the initial plume size, $S_p$. The simulations were done with an initial tracer concentration of 1000 μg m³.**

The concentration of OH within a real BB plume can be affected by many factors and is not directly determined by either the density or the age of the plume. For definiteness and simplicity, we assume that it does not depend on the initial parameters

of the plume and does not change as the plume evolves; based on the ambient measurements by Akagi et al. (2012), its value was set to $5\times10^6$ cm$^{-3}$ in all our simulations. We also assumed a constant temperature of 298 K, thus making unnecessary any assumptions regarding enthalpies of vaporization of SVOCs.

Along with aerosol species, MDMOA has been configured to simulate the evolution of an inert tracer. The tracer's initial mass concentration was the same as that of BB OA (that is, $C_0$). The behavior of the tracer's mass concentration is affected

only by dilution and was controlled by the parameter $S_p$ (initial plume size). Figure 1 shows the dynamics of the mass concentration of the inert tracer for several different values of $S_p$ according to our simulations. Consistent with the tracer dynamics demonstrated by Bian et al. (2017), our simulations indicate that the initial plume size may have a major impact on the subsequent plume evolution (and thus on the BB OA mass concentration). In particular, whereas the tracer concentration drops by four orders of magnitudes during the evolution of a plume with the smallest value of $S_p$ (100 m), it keeps a

practically constant value (decreasing by less than 3 %) within the largest plume considered (with $S_p$ of 100 km). Note that for the plumes with $S_p$ smaller than 3 km, the largest changes of tracer concentration occur during the first 5 hours of evolution; afterwards, the changes are relatively small and are not of significance for the results of this study. It should be emphasized that our simulations of the BB plume dispersion were not intended to be fully realistic and quantitatively





accurate. Rather, we used a simple plume dispersion model (which is formally not applicable at time scales exceeding a few hours and has many other limitations) just to specify several definite scenarios for the dilution process, with strongly different dilution rates during the initial stage of the BB plume evolution. Using the simulated tracer concentration ($Tr$), we evaluated the BB OA mass enhancement ratio (abbreviated as EnR throughout this paper and also denoted as $\gamma_a$ below) at a given time $t$ as follows:

$$\gamma_a(t) = \frac{C_{OA}(t) - C_{bg}}{Tr(t)},$$  (19)

where $C_{OA}(t)$ is the total OA concentration. Note that $\gamma_a$ is analogous to the "inert OA mass enhancement ratio" introduced by Bian et al. (2017). Since the initial concentration of the tracer was set to be the same as the initial concentration of BB OA, the initial value of $\gamma_a$ in our simulations was always equal to one without using any additional normalization (that is usually involved in similar definitions of EnR used in chamber and field studies). The simulations were performed with a sufficiently small nominal time step of 1 s. Based on some test simulations performed with smaller time steps, we estimate that the numerical error in $\gamma_a$ does not exceed 10 % in any case considered (but is typically much smaller).

Analysis of $\gamma_a$ allows us to identify changes of BB OA concentration due to the combined effects of oxidation and gas-particle partitioning processes. Furthermore, any kind of dependence of $\gamma_a$ on the initial BB plume size ($S_p$) or the initial concentration of BB OA ($C_0$) can be considered as a manifestation of a nonlinear behavior of the BB OA mass concentration. Indeed, it is easy to show using Eq. (6) that if the gas-phase concentration of each SVOC, $C_s^g$, were a linear function of the total concentration, $C_s^{tot}$, of the same species, then the dynamics of $\gamma_a$ would not depend on $C_0$ and could not be affected by dilution (and accordingly would not depend on $S_p$, either).

Interaction of SVOCs with water is not taken into account in the OA schemes described above; thus any known effects of humidity on evolution of BB OA, such as, e.g., formation of SOA from oxidation of water soluble organic compounds in the liquid phase (Brege et al., 2018), have been disregarded. However, as suggested by Tsimpidi et al. (2018), we used the calculations of the O:C ratio within the T18 and T18f schemes to characterize hygroscopicity and cloud condensation nucleus (CCN) activity of BB OA by means of the hygroscopicity parameter $\kappa$ (Petters and Kreidenweis, 2007). Specifically, we expressed the hygroscopicity parameter for any organic species $s$, $\kappa_s$, as a linear function of the O:C ratio by using a parameterization proposed by Lambe et al. (2011):

$$\kappa_s = 0{:}18\ (\text{O:C}) + 0.03.$$  (20)

The overall value of $\kappa$ for BB OA, $\kappa_{org}$, was obtained using a mixing rule (Petters and Kreidenweis, 2007):

$$\kappa_{org} = \sum_s \varepsilon_s \kappa_s,$$  (21)

where $\varepsilon_s$ is a volume fraction of a given species. The volume fractions were estimated by assuming a constant volumetric mass densities of $1.2 \times 10^3$ kg m$^{-3}$ for any OA species considered.


## 3 Results

### 3.1 Dynamical regimes of the BB OA evolution

The evolution of EnR ($\gamma_a$) according to our simulations performed with the different OA schemes is presented in Figs. 2 and 3. The simulations were done with several different values of the initial plume size, $S_p$, (see Fig. 2) and initial BB OA

concentration, $C_0$ (see Fig. 3). The parameter range considered in our simulations is intended to represent the highly variable characteristics of typical smoke plumes from any kind of vegetation fires. The simulation results allow us to identify five distinctive dynamical regimes of the BB OA evolution. The simulations corresponding to the specific regimes are marked in the figures by the numbers from 1 to 5. The first regime ("1") corresponds to a monotonic saturating increase of $\gamma_a$. This regime is found in the simulations with the C17, K15, and LIN schemes. An increase of $\gamma_a$ is followed by its decrease in the

second regime ("2") that is typical for the S15, T18 and T18f schemes. The third regime ("3") features a sharp initial decrease of $\gamma_a$, followed by a slow monotonic decrease. This regime is found with the C17, K15 and T18 schemes but only when $S_p$ is relatively small (in particular, when $S_p$ equals to 100 m, see Fig. 2c, d). The most complex behavior of $\gamma_a$ corresponds to the fourth regime ("4") and is found with the S15, T18, and T18f schemes (see Fig. 2c, d, Fig. 3d) for specific $C_0$ and $S_p$ values. In this regime, $\gamma_a$ first decreases, then increases and finally decreases again. Finally, the fifth regime ("5")

corresponds to the monotonic decrease of $\gamma_a$. It is found only in a simulation with the T18f scheme (see Fig. 3d).

The results shown in Figs. 2 and 3 demonstrate, on the one hand, that the differences in the considered representations of the OA evolution are associated with both major quantitative and qualitative differences in the BB OA behavior even when the control parameters are the same. For example, if $S_p = 5$ km and $C_0 = 10^3$ µg m$^3$ (see Fig. 2b), the simulation with the K15 scheme predicts a strong monotonic increase of EnR up to a factor of 3 after a 120-hour evolution. At the same time, the

T18f scheme predicts a slight net decrease of EnR, even though it also predicts an increase of EnR at the initial stage of the evolution. Even larger differences (exceeding a factor of 8) between the simulations are evident if $S_p$ equals to 0.1 km.

On the other hand, changing parameter values in the simulations with the same VBS scheme can result in "switching" between different types of the BB OA evolution. In particular, depending on $S_p$, the simulations with the C17 and K15 schemes can demonstrate both the regime "1" (Fig. 2a, b and Fig. 3) and the regime "3" (Fig. 2c, d). Both the regimes "2"

(e.g., Fig. 2a, b) and "4" (e.g., Fig. 2c) are found in the simulations performed with the S15 and T18f schemes. The simulations with the T18 scheme are found to manifest four regimes, from "1" to "4" (see Fig. 3a, c, d and Fig. 2c). In contrast to the simulations using the VBS framework, the simulations with the LIN scheme manifest a single dynamic regime (regime "1"). Note that the simulation results shown in Figs. 2 and 3 are not meant to provide an exhaustive analysis of the parameter space with regard to possible dynamical regimes but rather are intended to illustrate the diversity of the BB

OA behavior simulated with different OA schemes and different parameter values.





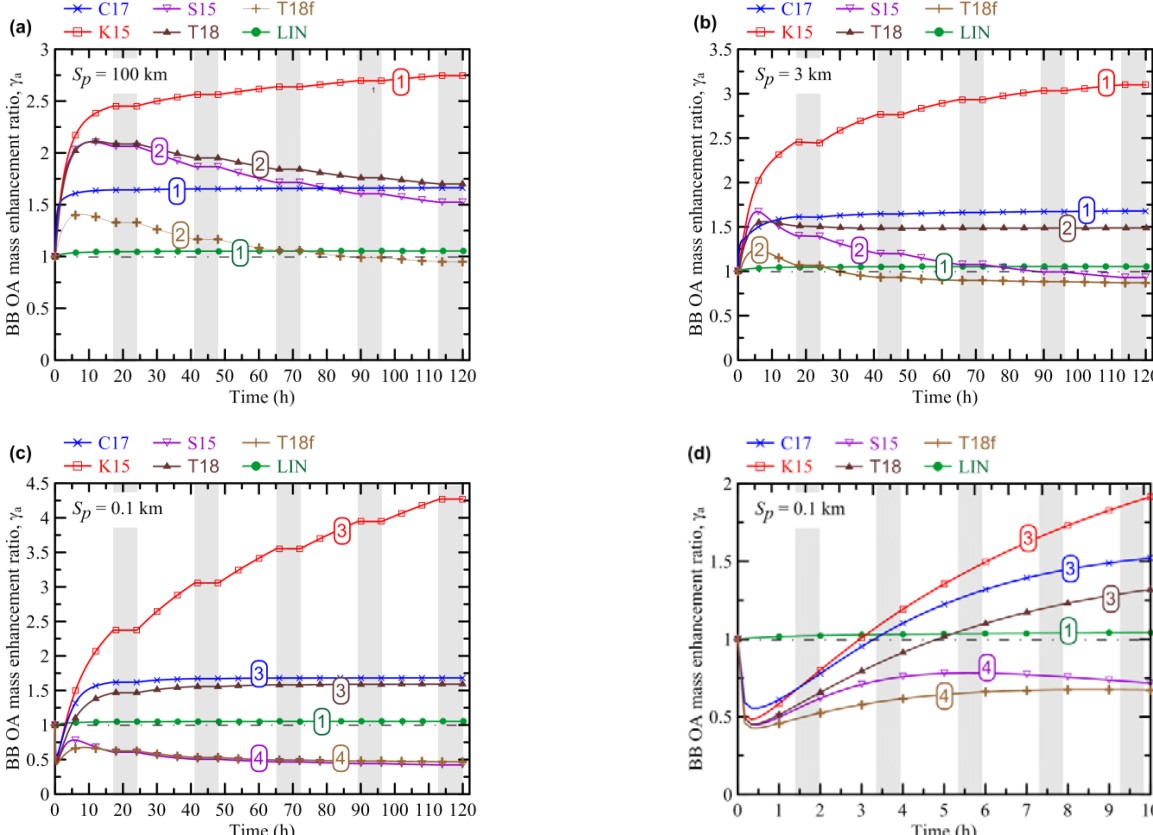

**Figure 2: Evolution of the BB OA mass enhancement ratio ($\gamma_a$) according to the simulations performed with the different OA schemes and with different values of the initial plume size: (a) $S_p = 100$ km, (b) $S_p = 3$ km, (c, d) $S_p = 0.1$ km. The initial BB OA concentration, $C_0$, was fixed at $10^3$ µg m$^3$ in all the simulations. Note that the plot (d) shows a zoomed fragment of the simulations presented in the plot (c). Shaded bands depict nighttime periods when no oxidation reactions were allowed to occur. The numbers on the curves denote the different dynamical regimes of BB OA evolution according to the definitions in Sect. 3.1.**

Figures 2 and 3 also show that using the same OA scheme with different values of $S_p$ and $C_0$ can be associated with not only qualitative differences in the EnR behavior (as indicated above) but also with considerable quantitative differences between the simulated EnR values. For example, $\gamma_a$ for the BB OA at an age of 120 hours that is obtained from the simulations with the K15 scheme and $C_0 = 1000$ µg m$^3$ increases from about 2.7 to 4.2 as $S_p$ decreases from 100 to 0.1 km (cf. Fig. 2a and 2c).

5   In contrast, the same change of $S_p$ in the simulation with the S15 scheme results in a dramatic drop of $\gamma_a$ from about 1.5 to 0.5. The simulations with the C17 and T18 are relatively insensitive to changes of $S_p$. An increase of $C_0$ is typically associated with a decrease of $\gamma_a$. The sensitivity of $\gamma_a$ to $C_0$ is very significant in the simulations with the K15 scheme but is weak in the simulations with the C15 and T18f schemes.





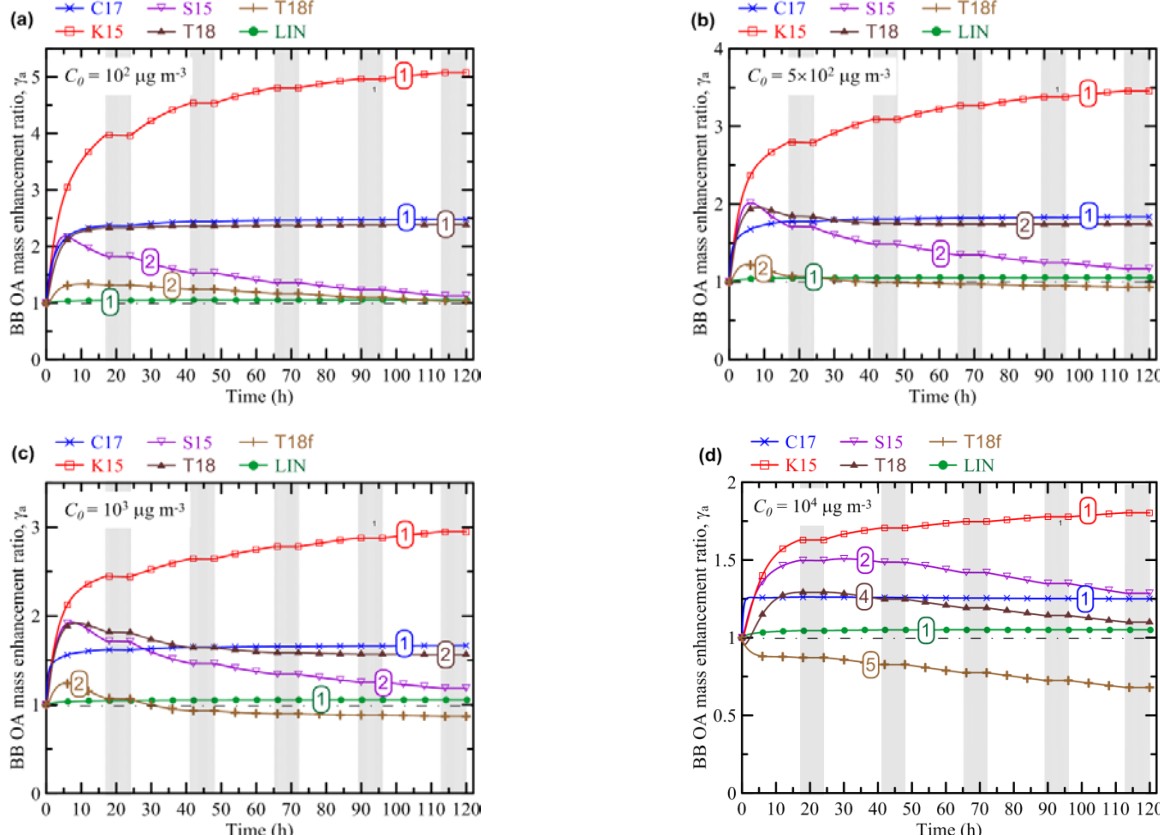

**Figure 3: Evolution of the BB OA mass enhancement ratio ($\gamma_a$) according to the simulations preformed with the different OA schemes and with different values of the initial mass concentration of BB OA: (a) $C_0 = 10^2$ µg m$^3$, (b) $C_0 = 5 \times 10^2$ µg m$^3$, (c) $C_0 = 10^3$ µg m$^3$, (d) $C_0 = 10^4$ µg m$^3$. The initial BB plume size, $S_p$, was fixed at 5 km in all the simulations shown.**

As noted in Sect. 2.3, any effects of changes in the parameters $S_p$ and $C_0$ on EnR can be considered as a manifestation of a nonlinear behavior of BB OA. In contrast to the nonlinear behavior demonstrated by BB OA in the simulations with the VBS schemes, the simulations with the LIN scheme manifest a simple "linear" behavior, being quite insensitive to changes of $S_p$ and $C_0$: $\gamma_a$ monotonically increases by ~5 % after 120 h irrespective of the parameter values. This insensitivity is a result of the assumptions that POA is not affected by any processes except for dilution and that the initial concentration of SOA precursors is proportional to $C_0$. The dependences of $\gamma_a$ on the control parameters of our simulations are analyzed in more detail in the next section (Sect. 3.2).





**(a)**                       **(b)**

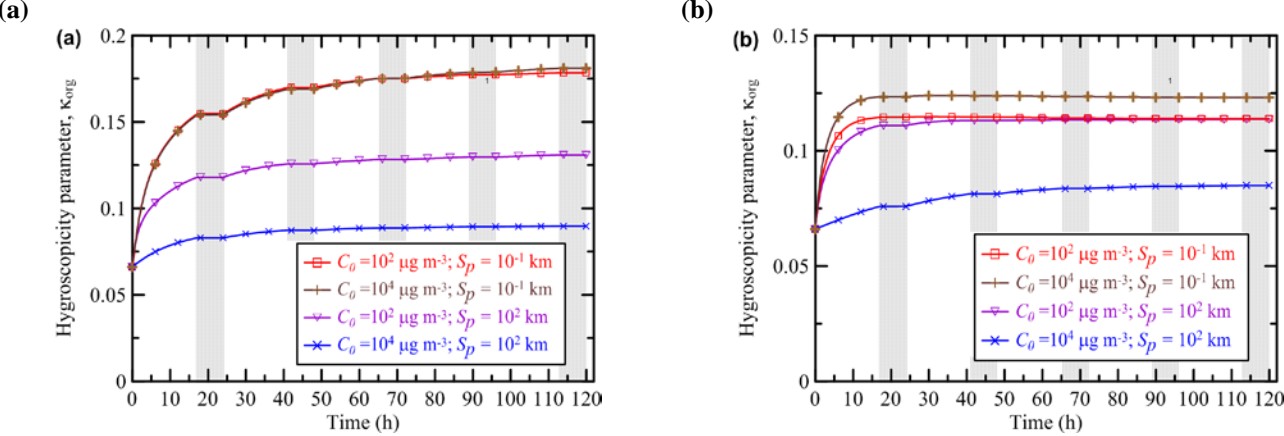

**Figure 4: Evolution of the hygroscopicity parameter κ simulated as a function of time with different values of the initial BB OA mass concentration, $C_0$ and the initial BB plume size, $S_p$, using the (a) T18 and (b) T18f schemes.**

The existence of the different dynamical regimes of BB OA evolution can be interpreted by considering the interplay between several competing processes, such as functionalization, fragmentation, and evaporation with dilution. In particular, regime "1" featuring a saturating enhancement of $\gamma_a$ can be explained by SOA formation dominated by functionalization reactions associated with a decrease of volatility and an increase of the molecular weight of SVOCs. A steady state is

reached in this regime when all SVOCs belong to the two lowest-volatility bins. Rapid fragmentation reactions in the case of S15 and T18f schemes or the decrease of the molecular weight of the products of the oxidation reactions (effectively combining the effects of the functionalization and fragmentation oxidation pathways) in the case of the T18 scheme can cause a depletion of the SOA amounts that have initially been formed from oxidation of POGs, leading to regime "2". A rapid POA evaporation caused by dilution at the initial stage of the BB plume evolution can result in a depletion of the bulk

OA amounts and give rise to the regimes "3", "4" or "5", depending on the interplay between fragmentation and functionalization after the fast dilution.

Unlike the BB OA enhancement ratio, the hygroscopicity parameter $\kappa_{org}$ exhibits, according to the simulations with the T18 and T18f schemes, a rather simple behavior (see Fig. 4), similar to that of $\gamma_a$ in the regime 1. The monotonic growth of $\kappa_{org}$ over time is hardly surprising, since each reaction is assumed to result in an increase of the O:C ratio (according to Eq. 14)

for at least one of the products with respect to that of the reactant (and the O:C ratio of the other products is assumed to remain the same as that of the reactant). The initial rapid increase of $\kappa_{org}$ is eventually slowing down because the semi-volatile POGs and SOGs participating in the oxidation processes are eventually transformed either into low-volatile products (as in the simulations with the T18 scheme) scheme or, on the contrary, into volatile gases (as in the simulations with the T18f scheme).



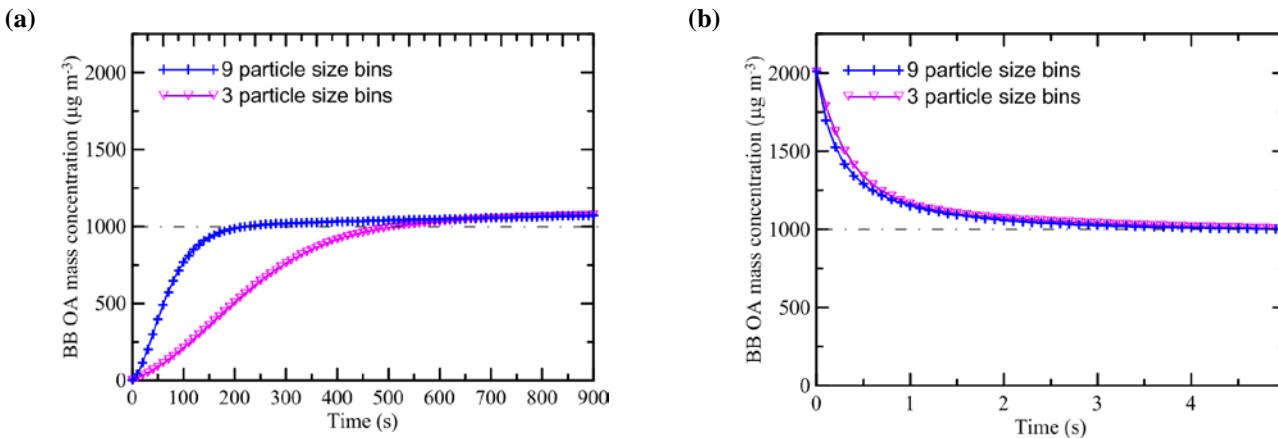

**Figure 5: Adjustment of mass concentration of fresh BB OA to thermodynamic equilibrium according to test simulations performed with the S15 scheme without dilution. The simulations were initiated from different initial conditions: (a) $C_0 = 0$ μg m³; (b) $C_0 = 2 \times 10^2$ μg m³. The thermodynamic equilibrium in the both cases corresponds to a BB OA concentration of $10^3$ μg m³. The time step in the test simulations was set at 0.1 s.**

Nonetheless, in spite a relatively simple behavior, the simulations of $\kappa_{org}$ are still rather sensitive to the parameters of the BB plume. In particular, the value of $\kappa_{org}$ after the 120-hour evolution is found, according to the simulation with the T18 scheme, to be about a factor of two bigger in the cases with a small plume ($S_p = 0.1$ km) irrespective of the initial aerosol concentration than in the cases with a large and dense plume ($S_p = 100$ km; $C_0 = 10^4$ μg m³). The sensitivity of the simulations

of $\kappa_{org}$ with the T18f scheme to the control parameter values is smaller but is still considerable. The fact that $\kappa_{org}$ is sensitive to the parameters of a BB plume can be regarded as another manifestation of the nonlinear behavior of BB OA: it is easy to show that in the absence of transfer of organic species between the gaseous and condensed phases of organic species, the oxidation rate (and, therefore, $\kappa_{org}$) would depend neither on $C_0$ nor $S_p$ (similar to the case of the simulations of $\gamma_a$ with the LIN scheme). Note that only simulations for the "extreme" values of $C_0$ and $S_p$ (among those considered in this study) are

shown in Fig 4. Simulations with other (intermediate) parameter values would fall between the brown and blue curves. The dependencies of $\kappa_{org}$ on $C_0$ and $S_p$ are analyzed in some more detail in the next section. Note also that quantitatively, the hygroscopicity parameter values ranging from 0.08 to 0.18 for aged aerosol in our simulations look rather reasonable. For example, Mikhailov et al. (2015) reported a volume-based hygroscopicity parameter value of about 0.1 (on average) for the accumulation mode of aerosol in Siberia during the period of intense fires.

As noted above (Sect. 2.3), the initial conditions for our BB OA simulations correspond to the instantaneous thermodynamic equilibrium. Additional test simulations performed using the S15 scheme with perturbed initial conditions and two different numbers of size bins (nine, as in the simulations with the C17, K15 and S15 schemes, or three, as in the simulations with the T18 and T18f schemes) confirm (see Fig. 5) that thermodynamic equilibrium between POGs and particles is typically





reached before any noticeable changes occur due to chemical processes. The equilibration time scales are found to range from just a few seconds (when the balance between gas and particle phases is shifted toward the particle phase) to about ten minutes (when all the BB emissions are assumed to be in the gas phase and the bin number equals three). In accordance with these results, the BB OA mass concentration after the five-day evolution was not found to be sensitive in our additional test

simulations (which are presented here) to either the initial perturbations of the gas-particle equilibrium or to the change in the number of bins in the particle size distribution in the range of the considered values of the plume parameters, even though aged as well as strongly diluted OA can feature much longer equilibration time scales (Riipenen et al., 2010).

### 3.2 Dependencies of the BB OA enhancement ratio and hygroscopicity parameter on the parameters of the smoke plume

Figures 6 and 7 present the simulated dependencies of EnR on the initial BB OA concentration, $C_0$, after 5 and 120 hours of evolution, $\gamma_a(5h)$ and $\gamma_a(120h)$, respectively. These dependencies have been calculated using the different OA schemes with varying values of the initial plume size, $S_p$. Figure 6 also shows (by dashed lines) the initial ratio of the total SVOCs mass concentration ($C^{tot}$) to the total BB OA mass concentration ($C_0$) at thermodynamic equilibrium according to Eq. (18). The $C^{tot}/C_0$ ratio depends on parameters of the assumed volatility distribution, and for this reason is different for different

schemes. Our simulations show that $\gamma_a(5h)$ follows, in most cases, a slow and monotonic inverse dependence on $C_0$. This dependence is qualitatively (and in some cases, but not always, even quantitatively) rather similar to that of $C^{tot}/C_0$: evidently, the decreasing character of the dependence of $\gamma_a$ on $C_0$ can be due to the fact that larger values of $C_0$ correspond to smaller SVOC gas-phase fraction available for SOA formation. Larger values of $S_p$ typically correspond to larger values of $\gamma_a(5h)$; such a dependence is evidently due to evaporation of POA with dilution. However, the dependences of $\gamma_a(5h)$ on

$C_0$ and $S_p$ are not always monotonic. In particular, the dependence of $\gamma_a$ on $C_0$ has a "weak" minimum in the simulations with the T18 scheme and $S_p$ of 1 km when $C_0$ is around $5\times10^3$ μg m$^3$. A similar minimum (but around $2\times10^3$ μg m$^3$ can be seen in the simulations with the T18f scheme). When $C_0$ equals to $5\times10^3$ or $10^4$ μg m$^3$ in the simulations with the T18 scheme, $\gamma_a$ calculated with $S_p = 5$ km is larger than that calculated with $S_p = 100$ km, although an inverse relation takes place between the corresponding values of $\gamma_a$ simulated with $S_p = 0.1$ km and $S_p = 0.5$ km. These irregularities cannot be easily explained but are

certainly not due to numerical errors (as has been confirmed in additional test simulations with a reduced time step), emphasizing the essentially nonlinear character of the BB OA behavior, especially when it is modeled using such complex VBS schemes as T18 or T18f.

A value of $\gamma_a(120h)$ also tends to decrease as $C_0$ increases in most cases, but with differences among the schemes : C17, K15 and T18 schemes with a weaker degree of fragmentation show a pronounced decrease, while schemes with a stronger degree

of fragmentation are more insensitive to on $C_0$ (see Fig. 7). This is presumably due to the fact that if functionalization is more important than fragmentation than the availability of more SOGs at lower $C_0$ leads to enhances SOA formation.





**Figure 6: Simulated dependencies of the BB OA enhancement ratio after 5 hours of evolution on the initial BB OA mass concentration, $C_0$. The dependencies are obtained with different OA schemes (see Table 1) and with different values of the initial plume size, $S_p$. Dashed lines show the ratio of the total initial mass concentration of SVOCs ($C^{tot}$) to the initial BB OA concentration ($C_0$) at thermodynamic equilibrium according to Eq. (18).**

However, different schemes features more complex and diverse dependencies on $S_p$ (see Fig. 7). For example, while the simulations with the C17 and T18 schemes, where the fragmentation and functionalization processes are treated implicitly, do not reveal any strong sensitivity of $\gamma_a(120)$ on $S_p$, the scheme K15 where functionalization strongly dominates over fragmentation yields a rather strong but inverse dependence of the same parameter (that is, unlike $\gamma_a(5h)$ calculated with the same scheme, $\gamma_a(120h)$ decreases with increasing $S_p$). Presumably, evaporation of POA due to dilution provides more "fuel"







**Figure 7: The same as in Fig. 6 but for the BB OA enhancement ratio after 120 hours of evolution and except that the ratio of $C^{tot}$ to $C_0$ (which is not supposed to characterize the state of BB OA after five-day aging) is not shown.**

for functionalization reactions, leading to a larger increase of $\gamma_a$ when $S_p$ is smaller (as found in the simulations with K15 scheme). In contrast, the schemes with strong fragmentation reactions, S15 and T18f, reveal a strong increasing dependence of $\gamma_a(120h)$ on $S_p$, while they are less sensitive on $C_0$ relatively to the other schemes. This dependence can mainly be due to the fact that the evaporated organic compounds are exposed to fragmentation reactions: accordingly, a decrease in $S_p$ results in larger losses of SVOCs and smaller SOA concentrations.

A comparison of the results shown in Fig. 6 and Fig. 7 reveals that the differences between the different VBS schemes are associated with mostly only quantitative differences in the dependencies of $\gamma_a(5h)$ on $C_0$ and $S_p$ but with both quantitative and



some qualitative differences in the corresponding dependencies of $\gamma_a$(120h). This observation indicates that the mass concentration of aged BB OA is likely to be much more strongly affected in the simulations by uncertainties in available representations of the BB OA evolution than the mass concentration of relatively fresh BB OA. One of the reasons is that fragmentation reactions become increasingly important with time when the SOG oxidation level increases, and then the competition between functionalization and fragmentation creates the more complex dependence on the plume parameters. One of the most significant findings of our analysis in this respect is that the sensitivity of $\gamma_a$ to the initial plume size (or, similarly, to the dilution rate) may be positive or negative, depending on a VBS scheme and the BB OA age. Factors controlling the dependence of $\gamma_a$ on $S_p$ are examined more in detail in the next section (Sect. 3.3).

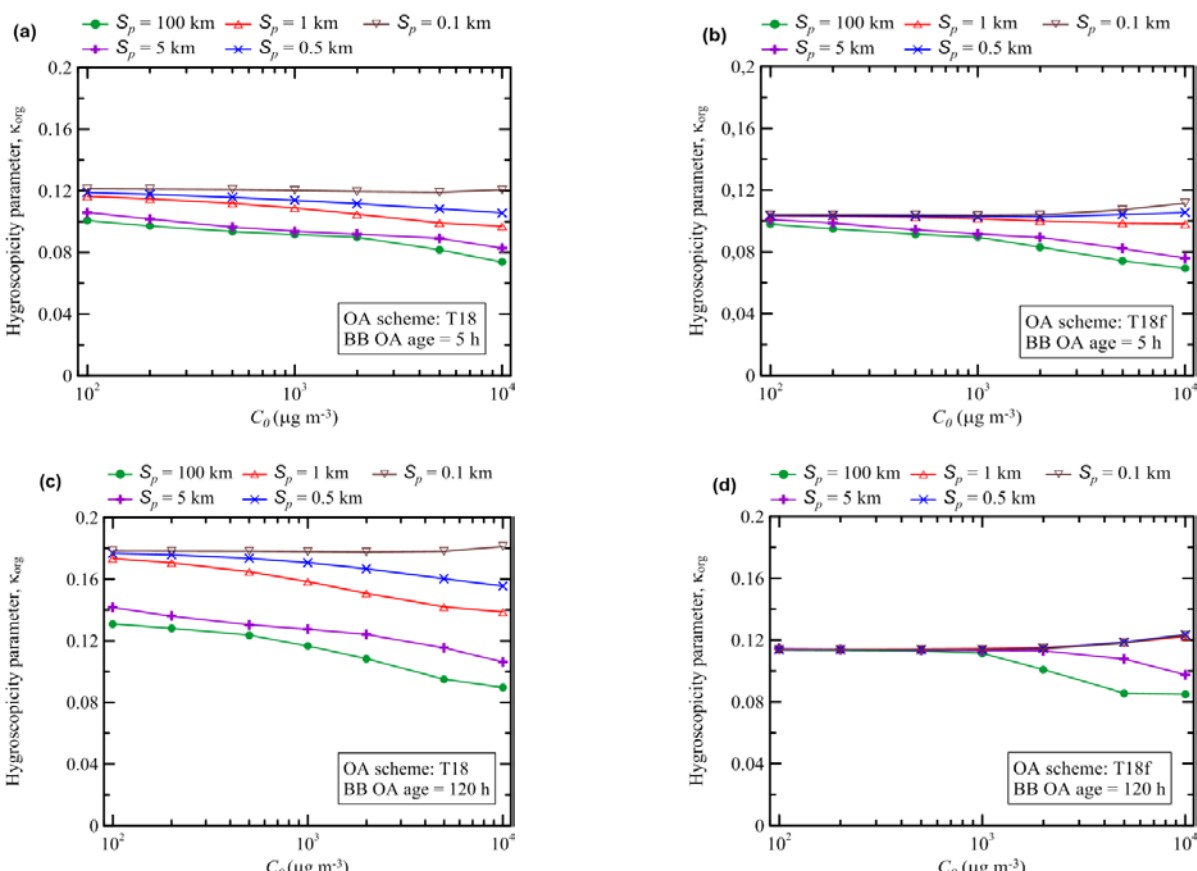

**Figure 8: Dependencies of the hygroscopicity parameter, $\kappa_{org}$, after (a, b) 5 hours and (c, d) 120 hours of BB OA evolution simulated using the (a, c) T18 and (b, d) T18f schemes.**

The simulated dependencies of the values of the hygroscopicity parameter after 5 and 120 hours of evolution, $\kappa_{org}$(5h) and $\kappa_{org}$(120h), respectively, on the initial BB OA concentration, $C_0$, and the plume size, $S_p$, are presented in Fig. 8. Evidently,





both $\kappa_{org}$(5h) and $\kappa_{org}$(120h) values are also affected by changes of the control parameters in a nonlinear way, although the manifestations of the nonlinearities are not as spectacular as in the case of the corresponding dependencies for EnR. In particular, while the values of $\kappa_{org}$(5h) and $\kappa_{org}$(120h) are decreasing with increasing $C_0$ for the largest plumes ($S_p = 100$ km), they are nearly constant and even slightly increasing with increasing $C_0$ for the smallest plumes ($S_p = 0.1$ km). According to

the simulations with both schemes, both $\kappa_{org}$(5h) and $\kappa_{org}$(120h) are more sensitive to the changes of $S_p$ than to the changes of $C_0$. Furthermore, the sensitivity of $\kappa_{org}$(5h) and $\kappa_{org}$(120h) to $S_p$ increases with increasing $C_0$. Note that as $C_0$ increases, a larger fraction of the total POM concentration, $C^{tot}$, belongs, under thermodynamic equilibrium, to the condensed phase. This fraction is oxidized (due to mass transfer between the particles and gas phase) slowly relative to the gas-phase fraction, which is why both $\kappa_{org}$(5h) and $\kappa_{org}$(120h) mostly tend to decrease with $C_0$. However, the particulate fraction of $C^{tot}$ can

readily evaporate and become accessible for fast gas-phase oxidation as a result of evaporation in the rapidly diluted small plumes: this may explain why $\kappa_{org}$(5h) and $\kappa_{org}$(120h) are nearly insensitive to $C_0$ in the simulations with $S_p = 0.1$ km.

The simulations of $\kappa_{org}$ performed with the T18 and T18f schemes are noticeably different both quantitatively and qualitatively. Quantitatively, the T18 scheme typically yields larger values of $\kappa_{org}$ than the T18f scheme. Qualitatively, while aerosol aging is associated with increasing sensitivity of $\kappa_{org}$ to $S_p$ in the simulations with the T18 scheme (cf. Fig. 8a and

Fig. 8c), the sensitivity of $\kappa_{org}$ simulated with the T18f scheme either diminishes or does not change significantly (depending on $C_0$) as the aerosol ages (cf. Fig. 8b and Fig. 8d). The likely reasons for the differences between the simulations of $\kappa_{org}$ with the two schemes have already been mentioned above (see Sect. 3.2). That is, on the one hand, oxidation reductions in the T18 scheme yield less volatile products which eventually accumulate in particles and increase the overall O:C ratio for BB OA. On the other hand, fragmentation reactions in the T18f scheme result in irreversible loss of the condensable organic

matter into the gas phase, thus limiting an increase of the O:C ratio for the condensed phase.

### 3.3 The effects of fragmentation reactions

As noted in the previous section, our simulations performed with the S15 (or T18f) and K15 schemes reveal qualitatively different dependencies of the BB OA enhancement ratio after 120 hours of evolution, $\gamma_a$(120h) on initial concentration $C_0$, but even more the initial plume size, $S_p$. Specifically, while the S15 and T18f schemes predict that $\gamma_a$(120h) increases with an

increase of $S_p$, the K15 scheme yields an inverse dependence. Meanwhile, the main difference between the S15 and K15 schemes is that the fragmentation pathway of the oxidation reactions has a larger weight with respect to the functionalization pathway in the S15 scheme, compared to the K15 scheme. The effect of fragmentation reactions is also well visible in simulations with the T18f scheme, when compared to the simulations with the T18 scheme. Taking these considerations into account, we performed additional simulations in which we varied the fragmentation branching ratio in the same scheme. For

definiteness, we used the S15 scheme which is rather "transparent" and flexible but yet not oversimplified.



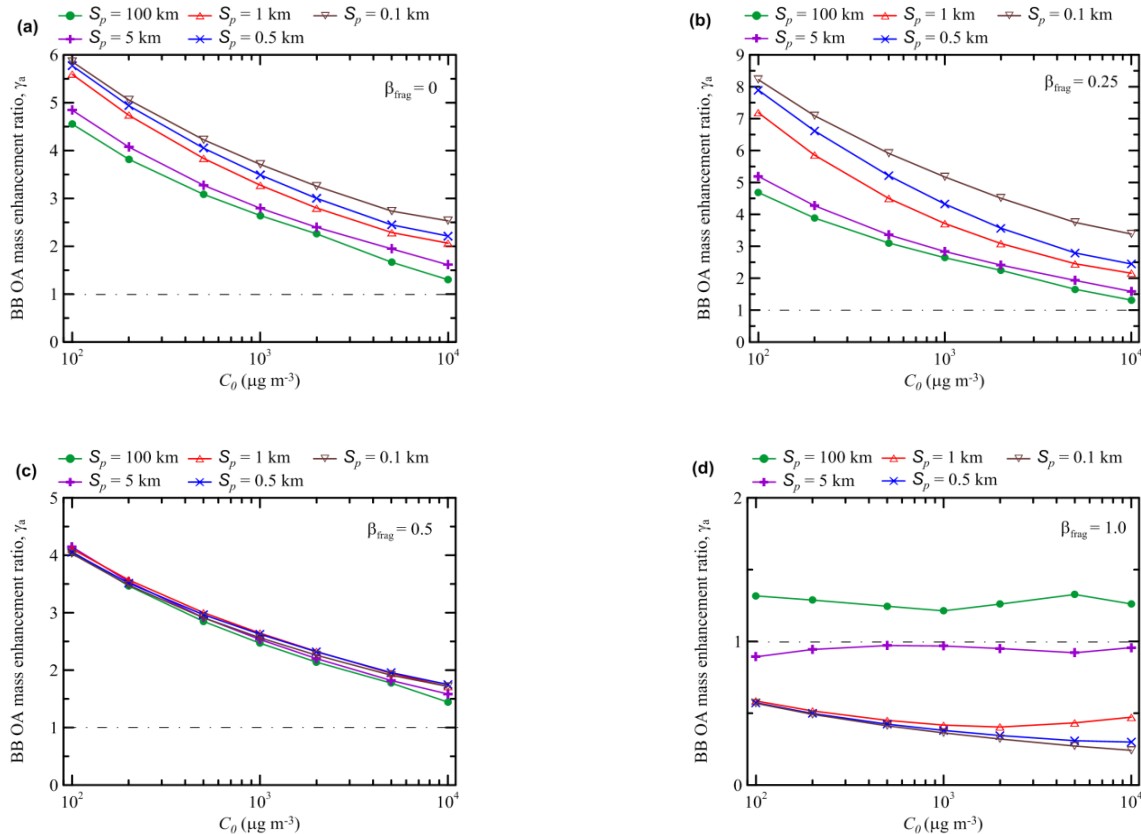

**Figure 9: Dependencies of the BB OA enhancement ratio after 120 hours of evolution on the initial BB OA mass concentration, $C_0$, according to the simulations using the S15 scheme with different values of $S_p$ and varying values of the fragmentation branching ratio: (a) $\beta_{frag} = 0$, (b) $\beta_{frag} = 0.25$, (c) $\beta_{frag} = 0.5$ and (d) $\beta_{frag} = 1.0$.**

Figure 9 shows the dependencies of $\gamma_a(120h)$ on both $C_0$ and $S_p$ according to our simulations performed with four different values of the fragmentation branching ratio, $\beta_{frag}$, for the SOA oxidation reactions (see Eq. 11) : 0, 0.25, 0.5, and 1.0. Figure 7c shows similar calculations corresponding to the case with $\beta_{frag} = 0.85$. It is noticeable that the dependencies obtained using the S15 scheme with $\beta_{frag}=0$ (see Fig. 9a) are very similar to those obtained with the K15 scheme (see Fig. 7b).

5    Specifically, according to the simulation with both schemes, $\gamma_a(120h)$ is inversely dependent on $S_p$. An increase of $\beta_{frag}$ (see Fig. 9b) to 0.25 results in the stronger sensitivity of $\gamma_a(120h)$ to $S_p$ and in an increase of maximum values of $\gamma_a(120h)$ (reached with $S_p=0.1$ km): apparently, this is due to the fact that the effect of fragmentation of SOGs is counteracted by oxidation of their products which are reprocessed according to Eq. (11). However, as $\beta_{frag}$ increases further, the reprocessing of SOGs cannot compensate for a loss of SVOCs in the fragmentation branch: the values of $\gamma_a(120h)$ calculated with

10    $\beta_{frag}=0.5$ (see Fig. 9c) are typically lower (and sometimes very considerably) than those calculated with $\beta_{frag} =0.25$.





Furthermore, when $\beta_{frag}$ equals 0.5, the dependence of $\gamma_a$(120h) on $S_p$ "collapses". That is, an approximate balance between the fragmentation and functionalization branches makes the BB OA enhancement ratio after 5 days of evolution nearly insensitive to dilution. Note that there is an obvious similarity between the dependences obtained using, on the one hand, the S15 scheme with the branching ratio of 0.5 (Fig. 9c), and, on the other hand, the C17 and T18 schemes. This similarity

implies that the fragmentation and functionalization processes are, effectively, nearly balanced in the C17 and T18 schemes. The dominance of fragmentation over functionalization is again associated with a rather strong sensitivity of $\gamma_a$(120h) to $S_p$ (see Fig. 7c and Fig. 9d). However, unlike the cases with $\beta_{frag}$ <0.5, $\gamma_a$(120h) increases with increasing $S_p$. Another noteworthy feature of the simulations with $\beta_{frag}$ =0.85 or $\beta_{frag}$ =1.0 is a rather weak sensitivity of $\gamma_a$(120h) to $C_0$. This is likely because the oxidation of POGs, which are available in larger amounts when $C_0$ is smaller, yields much less SOA in the cases

with dominating fragmentation, compared to the cases with dominating functionalization.

The effects of fragmentation and functionalization reactions on the composition of BB OA in the simulation performed with the S15 scheme on the composition of BB OA are further visualized in Fig. 10, which illustrates the BB OA evolution by distinguishing primary and secondary fractions in particles (POA and SOA) and in the gas phase (POG and SOG), as well as between "fresh" ($n{\leq}2$, see Sect. 2.2) and "aged" ($n{>}2$) fractions of SOA (SOA-f and SOA-a, respectively). The simulations

shown in Fig. 10 were performed for the four cases with different values of $C_0$ ($10^2$ and $10^3$ µg m$^{-3}$) and $\beta_{frag}$ (0 and 1). The initial plume size was taken to be 100 km in all these simulations; that is, the BB OA evolution was practically not affected by dilution.

It is evident that after about 5-10 hours of evolution, SOA already provides a major contribution to the OA mass concentration in all of the cases considered. Not surprisingly, the SOA fraction is larger in the simulations with $\beta_{frag}$ = 0 (in

the "functionalization" case) than in the simulations with the $\beta_{frag}$ = 1 (in the "fragmentation" case). In addition, the aged SOA fraction is larger for the non-fragmentation case. Consistent with the dependence of $\gamma_a$ on $C_0$ in Fig. 9a, the SOA fraction calculated with $\beta_{frag}$ = 0 is also larger in the simulations with a smaller value of $C_0$ (cf. Figs. 10a and 10b). A smaller $C_0$ value is also associated with a larger contribution of SOA-a (and correspondingly, with a smaller contribution of SOA-f) to the total concentration of SOA. It is worth noting that compared with the simulations performed with $C_0 = 10^3$ µg m$^{-3}$, the

simulations with $C_0 = 10^2$ µg m$^{-3}$ feature a faster decrease of SOA-f. This is probably due to an initially larger fraction of SOGs in the simulations with $C_0 = 10^2$ µg m$^{-3}$: rapid oxidation of "fresh" SOGs causes evaporation of corresponding species from particles, thus depleting SOA-f. A major qualitative difference between the simulations performed for the "functionalization" and "fragmentation" cases is that the OA, SOA, and SOA-a concentrations monotonically increase with time in the former case (regime 1) but demonstrate a "humped" dependence on time in the latter case (regime 2). That is, the

simulations shown in Fig. 10 confirm our initial suggestion (see Sect. 3.1) that realization of regimes "1" and "2" depends on the ratio between the fragmentation and functionalization branches of the oxidation reactions.





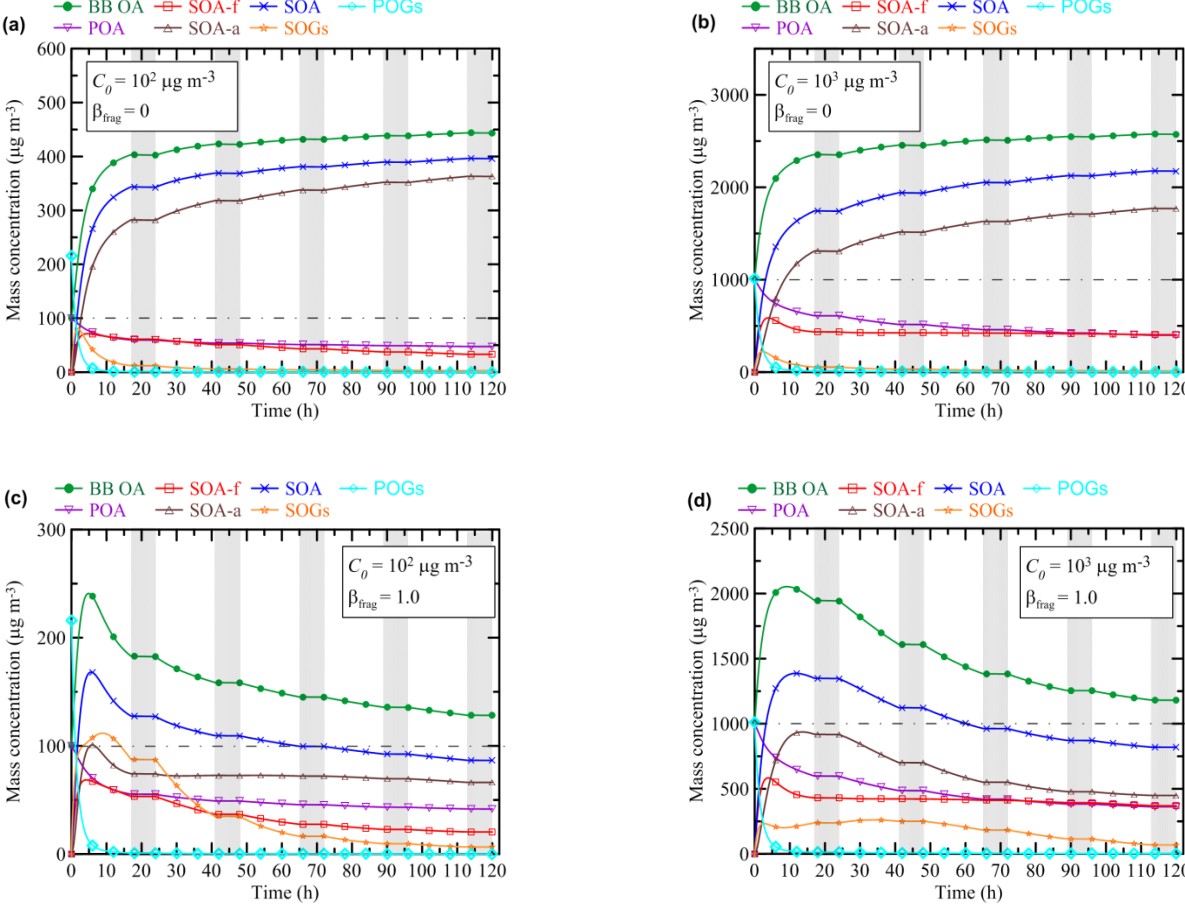

**Figure 10: Evolution of the mass concentration of BB OA and its components, including POA, "fresh" SOA (SOA-f), and "aged" SOA (SOA-a) according to the simulations performed with the S15 scheme. Also shown are the total mass concentration of SOA and mass concentrations of POGs and SOGs. The simulations were done with two different values of $\beta_{frag}$: (a, b) $\beta_{frag}$ = 0, (c, d) $\beta_{frag}$ = 1 and two different values of $C_0$: (a, c) $C_0$ = $10^2$ µg m$^3$, (b, d) $C_0$ = $10^3$ µg m$^3$. The initial plume size was set at 100 km in all the simulations.**

## 4 Discussion

In this section, we briefly discuss possible implications and limitations of our findings presented in Sect. 3. First of all, our results indicate that uncertainties associated with the representation of BB OA evolution in chemistry transport models (CTMs) may have a major impact on the simulated behavior of BB aerosol. Indeed, our simulations show that differences

5   between available OA schemes may entail differences in the calculated EnR, $\gamma_a$, as large as a factor of five (see Fig. 3a) or





even bigger (see Fig. 2c) after five days of evolution. Furthermore, there may be not only quantitative but also qualitative differences in the BB OA simulations, such as a difference between a gradually increasing $\gamma_a$ (in the regime "1") and a "humped" temporal dependence of $\gamma_a$ (in the regime "2") (see, e.g., Fig. 2a). Typically, OA VBS schemes are constrained (at least to some extent) with data from aerosol chamber experiments (e.g., Hennigan et al., 2011) which are, however, rarely

representative of more than a few hours of BB aerosol aging under atmospheric conditions. Taking into account that major quantitative and qualitative differences between our simulations with different OA schemes are manifested at longer timescales, our results therefore indicate the particular importance of observational constraints at the time scales ranging from tens of hours to several days. Potentially, such constraints can be provided by in situ aerosol measurements (e.g., Jolleys et al., 2012; Shrivastava et al., 2015) or remote (e.g., satellite) observations (Konovalov et al., 2017).

Even if one of the VBS schemes considered in this study were perfectly adequate, its application in the framework of a typical CTM would still likely entail considerable model biases due to several factors. First, the bias may be due to a dependence of BB OA evolution on its initial concentration. Indeed, smearing of BB emissions from an intense but small (relative to the grid cell size) fire within a model grid cell would result in smaller initial concentration of BB OA ($C_0$) in the model than in the real atmosphere. For example, if the model had a horizontal resolution of $50 \times 50$ km$^2$ but the fire size was

$5 \times 5$ km$^2$, the initial concentration of BB OA in the model would be two orders of magnitude lower than within the actual plume (e.g. $10^2$ instead of $10^4$ µg m$^{-3}$). As indicated by our analysis for several cases (see Fig. 6a, b, d and Fig. 7a, b, d), the EnR simulated for such a situation can be strongly overestimated due to its inverse dependence on $C_0$. Second, a model bias in EnR can be caused by a dependence of $\gamma_a$ on the initial plume size, $S_p$. For example, due to the effect of rapid dilution of BB OA plumes originating from relatively small (with a typical size of about 1 km or less) but multiple fires, a model that

does not resolve the plume scales will likely overestimate EnR and corresponding changes of the BB OA mass concentration as a result of its aging during several days, if the evolution of BB OA is strongly affected by fragmentation (see Fig. 7c, e), but it is also possible that a CTM will underestimate EnR, if functionalization dominates over fragmentation (see Fig. 7b). If the BB OA evolution is reproduced adequately with the T18 scheme, then the same situation may be associated with a negative bias in the hygroscopicity parameter $\kappa_{org}$ (see Fig. 8a) rather than with any significant bias in $\gamma_a$ (Fig. 7d). Net

effects of a limited horizontal resolution of a CTM on the simulated BB OA evolution are likely very variable and are hardly possible to accurately predict with a box model, as they should be dependent on the spatial structure and intensity of fire emissions as well as on meteorological condition. Third, the simulated values of $\gamma_a$ (and to less extent, $\kappa_{org}$) may be biased as a result of model errors in the vertical injection profile of fire emissions. Indeed, available parameterizations can strongly underestimate or overestimate the smoke injection height, sometimes by an order of magnitude (Sofiev et al., 2012); these

errors can cause biases in EnR as a result of the sensitivity of the simulations of BB OA to uncertainties in both the plume's density and dilution rate, and also to temperature affecting thermodynamic equilibrium. Similar effects of chemical nonlinearities on simulations of the consequences of atmospheric processing of BB emissions were first pointed out by Chatfield and Delany (1990).





The results of our analysis may be helpful for understanding the reasons for the observational diversity of BB OA aging effects. Indeed, our findings suggest that the occurrence of differences in the observed effects of BB OA aging, such as an increase (see, e.g., Yokelson et al. 2009; Konovalov et al., 2015; Vakkari et al., 2018) or a decrease (e.g., Jolleys et al., 2012; 2015) of EnR can be driven by differences in the initial parameters of the BB plumes, as well as by the BB aerosol's

photochemical age and the ratio between the fragmentation and functionalization reaction pathways. In particular, our simulations predict that aging of BB aerosol in a large plume is more likely to result initially in an increase of $\gamma_a$, whereas the initial evolution of a small plume is likely to be associated with an initial decrease of $\gamma_a$ (cf., e.g., Fig. 2a and Fig. 2c). Consistent with these predictions, there is observational evidence for a growth of $\gamma_a$ in the situations with large-scale fires (e.g., Konovalov et al., 2015; 2017; Vakkari et al., 2018) as well as evidence for an initial drop of $\gamma_a$ in a relatively small

isolated plume (Akagi et al., 2012). It is noteworthy that in agreement with our simulations corresponding to the regimes "3" and "4", the observations by Akagi et al. (2012) show an increase of $\gamma_a$ after the initial decrease. Our analysis also indicates that if fragmentation dominates over functionalization, an initial growth of $\gamma_a$ in situations with large fires is likely to be followed by a decreasing stage (see the simulations for regime "2" in Fig. 2a, b and Fig. 3). A similar "humped" dependence of $\gamma_a$ on the BB aerosol photochemical age was identified earlier in the analysis of satellite observations of BB

aerosol in Siberia (Konovalov et al., 2017). The fact that, according to aircraft observations, EnR can be around or smaller than unity even in relatively large plumes transported during a few days (Jolleys et al., 2012; 2015) is consistent with our simulations corresponding to the regimes "2", "4" or "5" in Fig. 3.

Our analysis indicates that BB OA evolution is strongly dependent on the fragmentation branching ratio (see Fig. 9) which is known to be a growing function of the oxidation state of OA (Jimenez et al., 2009). It seems reasonable to suggest that the

oxidation state of BB OA (and thus the effective fragmentation branching ratio) within a BB plume at any given moment is dependent on the initial chemical composition (and therefore oxidation state) of fresh BB OA. There is evidence (e.g., Grieshop et al., 2009; May et al., 2015; Tiita et al., 2016; Ahern et al., 2019; Lim et al., 2019) that the oxidation state of fresh BB OA is strongly variable, depending on conditions of burning and fuel type. Variable effects of fragmentation reactions have been suggested to be a possible reason for the diversity of values of the BB OA EnR (ranging from about 0.8 to almost

3) in the BB aerosol aging experiments (Hennigan et al. 2011). Therefore, the differences between our simulations performed with different oxidation schemes may reflect, to some extent, the diversity of the observed aging effects of BB OA due to variability of its initial chemical composition.

It should be especially noted that the VBS schemes employed in our simulations do not comprise the whole range of uncertainties associated with representation of BB OA aging in models. In particular, all of the considered schemes share the

common assumption that OA particles are composed of well-mixed liquids, which is also used in the majority of other modeling studies using the VBS framework. However, based on findings from some laboratory experiments, Shrivastava et al. (2013; 2015) argued that SOA should rather be represented as non-volatile glassy semi-solid mass. Available estimates of the annual-mean SOA glass transition temperature (Shiraiwa et al., 2017) indicate that in major biomass burning regions



(such as the Amazon basin and Siberia), SOA is typically in the liquid state near the ground but transits to the semi-glassy state towards the top of the boundary layer. Taking these estimates into account, it seems reasonable to expect that the SOA phase state in ambient BB aerosol in these regions is quite variable, depending on relative humidity and ambient temperature. When the equilibrium state of SOA is semi-solid, the effects of gas-phase fragmentation reactions are expected

to diminish (since SOA, once formed, cannot evaporate), depending on the highly uncertain time scale of transformation of semi-volatile SOA into a nonvolatile state (Shrivastava et al., 2015). Another uncertain factor that affects the evolution of the ambient BB aerosol but is not taken into account in our simulations is heterogeneous oxidation of OA particles. Similar to gas-phase fragmentation reactions, heterogeneous oxidation increases the oxidation state of particulate carbon, resulting in formation of volatile products escaping irreversibly to the gas phase. Kroll et al. (2015) estimated carbon loss from the

particles after one week of OA aging under typical atmospheric conditions to be in the range from 3 to 13%. Heterogeneous reactions would be slowed down further if the particles are in a semi-solid state (Shiraiwa et al., 2013). These estimates mean that heterogeneous oxidation is unlikely to affect BB OA behavior significantly at the time scales addressed in this study, except that including even a slow heterogeneous oxidation into our model would inevitably transform regime "1" (a monotonic saturating increase of $\gamma_a$, leading to a steady state) into regime "2" (an increase of $\gamma_a$, followed by its decrease), as

heterogeneous oxidation would eventually convert all particulate carbon into $CO_2$ (Donahue et al., 2013). Finally, it should again be noted that none of the VBS schemes considered takes into account interaction of SVOCs with water. That is, we essentially assume that both POA and SOA are formed of hydrophobic species. This assumption is not expected to have a strong effect on BB aerosol evolution in a relatively dry atmosphere (with the relative humidity below about 60 %) where the water uptake by BB aerosol is small (Hand et al., 2010). Significant uptake of water can result in important contributions of

aqueous phase oxidation reactions to transformations of BB OA chemical composition and to formation of SOA (Brege et al., 2018).

As argued above, the nonlinear behavior of BB OA in our model experiments stems merely from the well-established semi-volatile nature of organic compounds composing it. This fact indicates that the nonlinear features of BB OA evolution, which have been revealed in our simulations, are not a consequence of any simplifications involved in our model but rather

an inherent property of real BB aerosol. However, the great diversity of the simulations using different VBS schemes indicates that an accurate quantitative representation of these features in the presently available atmospheric models is yet hardly feasible. Taking into account that regional and global CTMs are not designed to address the scales associated with individual plumes, the results of our study indicate the need for robust subgrid parameterizations of BB OA evolution. Instead of explicitly assuming variable properties of individual smoke plumes and the BB aerosol they contain, such

parameterizations might rely on some external observable characteristics of fire emissions, such as the density and size distribution of fires spots and the fire radiative power, as well as on common meteorological parameters (e.g., temperature and relative humidity). Useful observational constraints for robust representations of atmospheric aging of BB aerosol in CTMs can hopefully be inferred from satellite measurements of BB aerosol (Konovalov et al., 2017).





## 5 Conclusions

In this paper, we analyzed the role of the intrinsic nonlinearity of the processes driving gas-particle partitioning and oxidation of SVOCs during the atmospheric evolution of BB organic aerosol. We performed simulations of BB OA evolution during a five-day period using a microphysical box model in which BB OA chemical transformations and SOA
formation were represented within the VBS framework. A simple parameterization based on a Gaussian dispersion model was used to specify several scenarios for dilution of a BB plume. The model was run with several VBS schemes of varying complexities, including 1D-, 1.5D-, quasi-2D- and 2D--VBS schemes that had been proposed in the literature to represent BB OA evolution specifically in the framework of regional and global chemistry transport models (CTMs). We analyzed the BB OA evolution by considering the BB OA mass enhancement ratio (EnR), $\gamma_a$, as a function of two control parameters, i.e.,
the initial horizontal plume size, $S_p$, across the wind direction and the initial BB OA concentration, $C_0$, corresponding to thermodynamic equilibrium. For a part of our simulations, we also considered the dependence of the hygroscopicity parameter, $\kappa_{org}$, on the same parameters. The initial plume size controls the dilution rate (the larger $S_p$, the smaller the dilution rate), and the initial BB OA concentration determines the partitioning of SVOCs between the gas phase and the particles (the larger $C_0$, the smaller the gas-phase fraction of SVOCs).

The simulation results allowed us to identify several qualitatively different regimes of BB OA evolution, which feature a monotonic saturating increase of $\gamma_a$ (regime "1"), increasing and then decreasing $\gamma_a$ (regime "2"), an increase of $\gamma_a$ after its initial rapid decrease (regime "3"), a stage with increasing $\gamma_a$ between two intermittent stages of its decrease (regime "4"), or monotonically decreasing $\gamma_a$ (regime "5"). The manifestations of nonlinear behavior of BB OA are found to include pronounced dependencies of $\gamma_a$ on both $S_p$ and $C_0$. For relatively fresh BB aerosol (with the age ranging from a few hours to
several tens of hours), EnR increases as $S_p$ increases or $C_0$ decreases. However, these kinds of dependencies can be strongly suppressed or even reversed, depending on the VBS scheme used and the aerosol age. Another interesting manifestation of nonlinear behavior of BB OA are possible shifts between the regimes as a result of a change in $S_p$: for example, the regimes "1" and "2" for slowly diluting large smoke plumes can transform into the regimes "3" and "4", respectively, for small plumes with fast initial dilution. Evolution of $\kappa_{org}$ is also found to be affected by changes of the control parameters in a
nonlinear way, although the manifestations of the nonlinearities are not as spectacular as in the case of the corresponding dependencies for EnR.

Differences between the VBS schemes can result in large quantitative differences between the simulations; they increase with the aerosol age and can almost be as large as a factor of 10 in the EnR values after a five-day evolution under typical conditions in summer mid-latitudes. Such large quantitative differences are usually associated with qualitative differences
between the simulations: specifically, the simulations resulting in the largest values of EnR correspond to regime "1" of BB OA evolution, while those yielding relatively small EnR values typically correspond to regimes "2", "4" or "5". Our analysis indicates that one of the major factors behind the quantitative and qualitative differences between the simulations with the different VBS schemes is the ratio between fragmentation and functionalization. Specifically, prevalence of fragmentation





over functionalization (when the effective fragmentation branching ratio exceeds 0.5) gives rise to regimes "2" and "4" and is associated with an eventual decrease of $\gamma_a$, while the dominance of functionalization over fragmentation is associated with regimes "1" and "3" that correspond to a saturating increase of $\gamma_a$. A change of the fragmentation branching ratio also can eventually cause a reversal of the dependence of $\gamma_a$ calculated after a five-day evolution on $S_p$: that is, $\gamma_a$ increases with

increasing $S_p$ if fragmentation is prevailing over functionalization and decreases otherwise.

We argue that the results of our study have important implications for modeling of BB OA in the framework of CTMs. First, our analysis allows us to point out nonlinear behavior of the OA system as a possible reason for the observed diversity of effects of aging of ambient BB aerosol (see, e.g., Cubison et al., 2011). A better understanding of the factors behind the diversity of BB OA aging effects is essential for ensuring the efficiency of in situ ambient observations of BB OA as

observational constraints to representations of BB OA processes in CTMs. Second, our findings suggest that uncertainties associated with representation of BB OA in CTMs may have a major impact on the simulated behavior of BB aerosol at the scales of its typical lifetime in the boundary layer with respect to dry deposition. Note that these uncertainties can be especially important in the context of modeling rapid climate change in the Arctic, where BB OA provides a considerable contribution to the radiative balance (Sand et al., 2015). Third, a rather strong sensitivity of EnR evolution to the parameters

of a BB plume (such as $C_0$ and $S_p$) indicates that application of even a perfect VBS scheme in the framework of available regional or global models would likely entail considerable model biases in simulations of atmospheric transformations of BB aerosol due to the fact that the evolution of individual BB plumes with varying parameters cannot be accurately represented on a typical model grid with a horizontal resolution ranging from tens to a few hundreds of kilometers. Overall, our findings call for the development of subgrid parameterizations of the BB OA evolution, which could be constrained with available in

situ and satellite measurements but, at the same time, would be sufficiently robust with respect to nonlinear effects that cannot be properly addressed in typical CTMs.

***Data availability.*** The data presented in this paper were obtained by integrating the dynamic equations specified in Sect. 2.1 and are available upon request from the corresponding author. Numerical codes representing the equations and the involved chemical and microphysical processes in the framework of the box model used in this study are based on the routines and

interfaces of the CHIMERE chemistry transport model. The CHIMERE codes are available at http://www.lmd.polytechnique.fr/chimere/, last access: 23 April 2019.

***Author contributions.*** IBK and MB designed the study. IBK also performed the simulations and data analysis and wrote the paper. NAG contributed to development of the box model. MOA contributed to the discussion of the results and to writing the paper.

***Competing interests.*** The authors declare that they have no conflict of interest.



***Acknowledgements.*** The simulations and analysis of evolution of the BB OA mass enhancement ratio were supported by the Russian Foundation for Basic Research (grant no. 18-05-00911). The simulations of the hygroscopicity parameter were performed with support from the Russia Science Foundation (grant agreement no. 18-17-00076). I.B. Konovalov acknowledges travel expenses in the framework of the French PARCS (Pollution in the ARCtic System - PARCS) project.

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
