# Peer review of "Nonlinear behavior of organic aerosol in biomass burning plumes: a microphysical model analysis"

_Atmospheric Chemistry and Physics, 2019_

## Referee Comment (RC1) · Anonymous Referee #2 · 27 Jun 2019

**Summary/recommendation:**

This is an excellent paper that provides a simplified overview of the anticipated effects on mass enhancement and hygroscopicity of plume size, initial mass loading, and assumed functionalization and fragmentation reactions. The authors provide details on both near-field (5 hours of aging) and highly aged (120 hours of aging) results. Both timescales are of interest to the biomass burning aerosol community. The authors have done a good job carefully walking through their many results. They also provide an excellent 'discussion' section in which they place their work within the broader framework of observations and chemical transport models; this section should serve as a nice reference for many in the community and will hopefully assist in guiding others with further lines of research based on the authors' work. I recommend that this paper be published after a few minor revisions, as stated below.

**General Comments:**

Part of the basis of this study is to build upon the Bian et al. (2017) model. Hodshire et al. (2019) also built upon the Bian et al. (2017) model to test in a similarly theoretical model the effects of fire (plume) size and background aerosol concentration on the near-field aging of aerosol size distributions. This paper may be worth mentioning within the intro and may provide as an appropriate citation for portions of the near-field discussions of EnR.

Hodshire, A. L., Bian, Q., Ramnarine, E., Lonsdale, C. R., Alvarado, M. J., Kreidenweis, S. M., Jathar, S. H. and Pierce, J. R.: More than emissions and chemistry: Fire size, dilution, and background aerosol also greatly influence near-field biomass burning aerosol aging, J. Geophys. Res. Atmos., 2018JD029674, doi:10.1029/2018JD029674, 2019.

Page 9: In Table 1 for the S15 scheme, the authors reference '"fresh" SOG ($\beta$frag = 0) and reactions involving "aged" SOG ($\beta$frag = 0.85)': it's clear where $\beta$frag = 0.85 come in from the text on pg 9 and equation 11. However, it's not clear where $\beta$frag = 0 would be in equations 9 or 10, if one where to assume that $\beta$frag != 0 for the fresh SOG. Can the authors include this information within the relevant equations?

Page 10: Can the authors comment in the text on why they chose to only use the FragSVSOA configuration?

Page 13: "This period is representative of the typical lifetime of BB aerosol in Siberia under conditions without precipitation (Paris et al., 2009)": can the authors comment on the relative lifetime of BB aerosol in other important fire environments, such as the Amazon, Africa, etc, in order to place the 120 hour designation into a broader context? Also, as the authors are choosing variables representative of Siberia (e.g. diurnal cycle) chose to use 5 ug m-3 as their background aerosol concentration, can they comment on how well this is anticipated to represent the Siberian

natural background during the fire season? The authors may also consider pointing out that only considering a relatively clean background is a limitation of the study, as entrainment of more polluted backgrounds will change the partitioning and evaporation rates of the plume particles (see e.g. Hodshire et al., 2019).

Hodshire, A. L., Bian, Q., Ramnarine, E., Lonsdale, C. R., Alvarado, M. J., Kreidenweis, S. M., Jathar, S. H. and Pierce, J. R.: More than emissions and chemistry: Fire size, dilution, and background aerosol also greatly influence near-field biomass burning aerosol aging, J. Geophys. Res. Atmos., 2018JD029674, doi:10.1029/2018JD029674, 2019.

Page 15: The authors could consider using only the 'EnR' or the '$\gamma_a$' notation throughout, as having 2 different variable names for the same thing is a little confusing.

Section 3.1: Can the authors briefly justify in the text why they chose their given fixed value of initial mass loading for the analysis in Fig 2 and fixed value of plume size for the analysis in Fig 3?

Also, why does Figure 4 (and Figure 8) only show results for the T18 and T18f schemes? Perhaps along with a brief justification for this choice the authors could also consider including the results of the other VBS schemes in a supplemental figure. I see that the authors note on lines 9-10 of page 20 "Note that only simulations for the "extreme" values of C0 and Sp (among those considered in this study) are shown in Fig 4. Simulations with other (intermediate) parameter values would fall between the brown and blue curves." But does this include all other schemes, that is, that all of the VBS schemes used fall between the brown and blue curves? This should be made clear in the text.

The authors could help the reader by being more explicit as to why smaller fires have higher hygroscopicity parameter values than larger fires. A brief sentence or 2 would go well on pg 20 (first paragraph) that clearly points out again that smaller fires can undergo more oxidation reactions (more gas phase material available both from initial partitioning and initial evaporation by dilution). Plume-size dependencancy on e.g. oxidation and partitioning is a complex subject that could aid from simple explanations and reminders such as this throughout.

Page 26: Can the authors comment in the text on whether there is information to say how realistic each βfrag value is?

It is worth pointing out somewhere within the text (possibly in the methods?) that a limitation of this study is that the largest fires may be have limited oxidation reactions and thus limited SOA

formation and/or fragmentation occurring within the dense initial plumes if the plume is dense enough to limit photochemistry, and that this study doesn't try to account for that effect.

**Figures/Tables:**

For each figure that has a horizontal dashed line at EnR=1, the caption should state that that line is there to indicate where 'no mass enhancement' occurs.

Figure 2: What are the shaded bands on panel d? Presumably this is not a designation of 'nighttime'. They may be present to guide the eye but are confusing and should be explained. I recommend either removing them or making them distinct from the nighttime bands in panels a-c.

Figure 6 and 7: It's difficult to see the dashed line for $C^{Tot}/C_o$ within some of the panels (e.g. Fig 6 panel a); can this line be made more distinctive?

**Technical comments:**

Page 5 line 11: Bian et al., 2017 simulated 4 hours of aging (not 5).
Page 8 line 17: are the SVOCs evenly distributed across the 5 bins? Please clarify

---

## Referee Comment (RC2) · Anonymous Referee #1 · 1 Jul 2019

Review of "Nonlinear behavior of organic aerosol in biomass burning plumes: a microphysical model analysis"

Summary: This manuscript addresses the different observed effects of atmospheric aging of organic aerosol in biomass burning plumes and their lack of representation in models. The authors used different complexities of volatility basis set (VBS) schemes in a microphysical dynamic model to simulate biomass burning organic aerosol evolution. They analyzed the nonlinear governing processes that drive the evolution of biomass burning organic aerosols. They also considered the biomass burning organic aerosol mass enhancement ratio, examined the behavior of the hygroscopicity param-

eter, and identified five different regimes of organic aerosol evolution. they come to the conclusion that the evolution depends on initial concentration and plume size of the specific biomass burning smoke plume, and such parameters are not usually captured in models.

This manuscript is very well written, insightful, in the scope of this journal, and will attract community attention for its significance. I appreciate that in the text, the authors not only emphasized the importance of this work but also provided some insights into associated uncertainties (and/or assumptions) and hinted at areas of improvements for future studies. I recommend it for publication in ACP, and I have the following minor suggestions for improvements.

Text comments: The use of chemistry transport models to simulate a single plume seems to be off-scale. Would the authors comment on the uncertainties associated with this disadvantage?

Page 10: Would the authors comment on why they did not include NVSOA formation in MDMOA? Or is it the conventional OA scheme that was mentioned on page 8 line 6?

Page 13 line 24-26: Would the authors specify in the text each of the size bin's range? Also, which 3 size bins were used for T18 and T18f?

Page 14 lines 3-7: "The concentration of OH ....based on the ambient measurements by Akagi et al. (2012), its value was set to $5\times106$ cm-3 in all our simulations. We also assumed a constant temperature of 298 K... " Is a plume environment equivalent to ambient conditions? e.g. is it valid to assume a constant temperature of 298K and ambient OH concentration in a plume? Would limited photochemistry within a plume reduce OH concentration?

Page 14 lines 8: "Along with aerosol species, MDMOA has been configured to simulate the evolution of an inert tracer. " Would the authors please clarify the concept and use of an inert tracer (it was initially mentioned in the abstract)? What is its composition

and properties?

Page 19, Figure 4, perhaps setting both figures with the same y-axis scale would be helpful. Also, in the text, it's unclear why only T18 and T18f schemes are shown, but not other schemes.

Page 24 "This observation indicates that the mass concentration of aged BB OA is likely to be much more strongly affected in the simulations by uncertainties in available representations of the BB OA evolution than the mass concentration of relatively fresh BB OA. One of the reasons is that fragmentation reactions become increasingly important with time when the SOG oxidation level increases, and then the competition between functionalization and fragmentation creates the more complex dependence on the plume parameters." Would the authors please elaborate on or quantify the competition between functionalization and fragmentation? e.g.the branching ratios?

Page 32 lines 28-29: The authors mentioned that such differences among VBS are under the "typical conditions in summer mid-latitudes", it would certainly be intriguing for future studies to examine how these behaviors vary in different environments.

Figures comments: It's unclear what the shaded greys represent in the figures, and they are quite distracting, I suggest that the authors justify them in the caption or remove them.

It may be already sufficient that figures have different colored lines. Adding different symbols are just adding noise to the figures (just as with the grey shades). But it could be a personal preference, just a suggestion.

Again, it would be helpful to compare different schemes if the axis scales are the same when possible (in the same magnitude range).

For consistency among figures, I would suggest that Figure 4 follow other figures (Fig 3 etc) to include legends in the same locations.

Figure 6's (and Figure 7's) caption mentions "dashed lines," although at first read (without looking at the legend's Ctot/C0), it's unclear if it's the dashed lines with dots and dashes, or dashes alone (which should be Ctot/C0). There two types of dashes, it'd be helpful to distinguish the two.

[Figure]

---

## Author Comment (AC1) · 25 Aug 2019

Referee's comment: *Part of the basis of this study is to build upon the Bian et al. (2017) model. Hodshire et al. (2019) also built upon the Bian et al. (2017) model to test in a similarly theoretical model the effects of fire (plume) size and background aerosol concentration on the near-field aging of aerosol size distributions. This paper may be worth mentioning within the intro and may provide as an appropriate citation for portions of the near-field discussions of EnR.*

We thank the referee for pointing out this omission: unfortunately, we were not aware of the very recent paper by Hodshire et al. prior to completion of our study and submission of our manuscript to ACP. The corresponding reference is provided in the revised manuscript, and the Hodshire et al. (2019) paper is mentioned there several times in various contexts. In particular, we note (in the Introduction) that Hodshire et al. (2019) pointed out a significant impact of background aerosol on near-field BB OA aging processes, and (in Sect. 4) that our findings concerning the impact of the plume size on EnR after a few initial hours of aging are qualitatively consistent with the results of numerical experiments conducted by Bian et al. (2017) and Hodshire et al. (2019). We also tried to make it clear (in Sect 2.3) that the configuration of the numerical experiments in our study is largely similar to that in both Bian et al. (2017) and Hodshire et al. (2019).

Referee's comment: *Page 9: In Table 1 for the S15 scheme, the authors reference '"fresh" SOG ($\beta_{frag} = 0$) and reactions involving "aged" SOG ($\beta_{frag} = 0.85$)': it's clear where $\beta_{frag} = 0.85$ come in from the text on pg 9 and equation 11. However, it's not clear where $\beta_{frag} = 0$ would be in equations 9 or 10, if one to assume that $\beta_{frag} = 0$ for the fresh SOG. Can the authors include this information within the relevant equations?*

We are sorry for this minor textual inconsistency. To address it, the description of the S15 scheme in Table 1 has been revised: we tried to make clear that a specific value of the fragmentation branching ratio is applicable only to oxidation reactions involving "aged" SOGs, while POGs and "fresh" SOGs are assumed to be not affected by fragmentation at all.

Referee's comment: *Page 10: Can the authors comment in the text on why they chose to only use the FragSVSOA configuration?*

A corresponding comment is provided in the revised manuscript. In particular, we note that the FragSVSOA configuration enables better consistency of the S15 scheme with the other VBS schemes considered in our study, and thus any differences between the simulations performed with the S15 scheme and the other schemes are easier to interpret. We also note that possible formation of NVSOA due to particle-phase reactions is among the factors (discussed more in detail in Sect. 4) that can affect the real BB OA evolution but that were not analyzed in our study, as it is focused on identification of major qualitative nonlinear effects in the BB OA behavior due to gas-phase oxidation reactions in BB plumes.

Referee's comment: *Page 13: "This period is representative of the typical lifetime of BB aerosol in Siberia under conditions without precipitation (Paris et al., 2009)": can the authors comment on the relative lifetime of BB aerosol in other important fire environments, such as the Amazon, Africa, etc, in order to place the 120 hour designation into a broader context? Also, as the authors are choosing variables representative of Siberia (e.g. diurnal cycle) chose to use 5 ug m⁻³ as their background aerosol concentration, can they comment on how well this is anticipated to represent the Siberian natural background during the fire season? The authors may also consider pointing out that only considering a relatively clean background is a limitation of the study, as entrainment of more polluted backgrounds will change the partitioning and evaporation rates of the plume particles (see e.g. Hodshire et al., 2019).*

In response to the Referee's comments, we have revised the second sentence of Sect. 2.3. We point out that the period of 120 hours has been chosen to be within the range of typical atmospheric lifetimes of submicron aerosol particles emitted from open vegetation fires in the major BB regions worldwide, as indicated, e.g., by a measurement-based estimate (5.1 days) of the lifetime of black carbon (BC) in Siberia (Paris et al., 2009) and global-model estimates (Wang et al., 2016) of the BC lifetimes for open fires in northern Africa (5.6 days) and northern South America (3.1 days).

In respect to our choice of the background OA concentration of *5* μg m$^{-3}$, we note that the same value was specified in the box model simulations performed by Bian et al. (2017) and that, for comparison, particulate matter ($PM_{10}$) in a boreal environment of central Siberia under background conditions (that is, without the detectable influence of local or regional pollution sources, including fires) was found by Mikhailov et al. (2017) to have concentrations ranging from about 2 to 10 μg m$^{-3}$ in summer, being composed mostly of organic material. Finally, we also note that specifying a much larger or much smaller value of the background OA concentration would likely result in noticeable quantitative changes of the simulated BB OA behavior, since entrainment of background aerosol affects evaporation rates and gas-particle partitioning in a BB plume (Hodshire et al., 2019).

Referee's comment: *Page 15: The authors could consider using only the 'EnR' or the '$\gamma_a$' notation throughout, as having 2 different variable names for the same thing is a little confusing.*

We have carefully considered the Referee's suggestion. We would like to note that as explained in the reviewed manuscript before Eq. (19), "EnR" is introduced as an abbreviation (rather than a notation) for the "enhancement ratio". A corresponding notation ($\gamma_a$) was introduced in Eq. (19) to allow us to present our quantitative results in a concise way. We hope that using both an abbreviation and a mathematical notation for the same physical characteristic is consistent with the standards of ACP. Nonetheless, to enhance the readability of the text, we tried to avoid (or at least, to minimize) using both $\gamma_a$ and EnR in the same section of the revised manuscript. More specifically, the use of "$\gamma_a$" is predominately reserved for presentation of quantitative results of our simulations in Sect. 3, while "EnR" is mostly used to discuss qualitative implications of our findings in Sects. 4 and 5. We hope that in this way a possible confusion between "EnR" and "$\gamma_a$" has generally been avoided.

Referee's comment: *Section 3.1: Can the authors briefly justify in the text why they chose their given fixed value of initial mass loading for the analysis in Fig 2 and fixed value of plume size for the analysis in Fig. 3?*

A corresponding brief explanation in introduced in the first paragraph of Sect. 3.1 of the revised manuscript. Specifically, we note that the fixed values of $C_0$ ($10^3$ μg m$^3$) and $S_p$ (5 km) in the simulations shown in, respectively, Fig. 2 and Fig. 3 are chosen to approximately represent mid-range values of the corresponding parameters (on a logarithmic scale).

Referee's comment: *Also, why does Figure 4 (and Figure 8) only show results for the T18 and T18f schemes? Perhaps along with a brief justification for this choice the authors could also consider including the results of the other VBS schemes in a supplemental figure. I see that the authors note on lines 9-10 of page 20 "Note that only simulations for the "extreme" values of C0 and Sp (among those considered in this study) are shown in Fig 4. Simulations with other (intermediate) parameter values would fall between the brown and blue curves." But does this include all other schemes, that is, that all of the VBS schemes used fall between the brown and blue curves? This should be made clear in the text.*

Following the Referee's suggestion, results of our simulations with the C17 scheme have also been included in Figures 4 and 8 of the revised manuscript. In addition, we explain (both in Sect. 2.3 and Sect. 3.1) that the hygroscopicity parameter was calculated only with the C17, T18 and T18f schemes because the other oxidation schemes (K15, S17 and LIN) considered in our study are not designed to evaluate the O:C ratio. We also tried to make clear that in the text fragment cited by the Referee, we mean the simulations performed using a particular scheme corresponding to each plot in Fig. 4.

Referee's comment: *The authors could help the reader by being more explicit as to why smaller fires have higher hygroscopicity parameter values than larger fires. A brief sentence or 2 would go well on pg 20 (first paragraph) that clearly points out again that smaller fires can undergo more oxidation reactions (more gas phase material available both from initial partitioning and initial evaporation by dilution). Plume-size dependency on e.g. oxidation and partitioning is a complex subject that could aid from simple explanations and reminders such as this throughout.*

Following the Referee's suggestion, a corresponding explanation is introduced in Sect. 3.1 of the revised manuscript.

Referee's comment: *Page 26: Can the authors comment in the text on whether there is information to say how realistic each $\beta_{frag}$ value is?*

Unfortunately, we are not aware of any strong experimental or theoretical constraints to the $\beta_{frag}$ values involved in the BB OA oxidation schemes. To address the Referee's comment, we note in the revised manuscript that simulations with the extreme values of $\beta_{frag}$ (0 and 1) can hardly correspond to any realistic situation and are provided only for reference purposes, but taking into account that the fragmentation branching ratio is a highly uncertain parameter of VBS schemes, neither of the simulations with the other values of $\beta_{frag}$ is intended to be more or less realistic.

Referee's comment: *It is worth pointing out somewhere within the text (possibly in the methods?) that a limitation of this study is that the largest fires may be have limited oxidation reactions and thus limited SOA formation and/or fragmentation occurring within the dense initial plumes if the plume is dense enough to limit photochemistry, and that this study doesn't try to account for that effect.*

The suggested comment (with an appropriate reference to Konovalov et al. (2016) where the effect mentioned by the Referee was addressed using a 3D model) is included in Sect. 2.3 of the revised manuscript.

Referee's comment: *For each figure that has a horizontal dashed line at EnR=1, the caption should state that that line is there to indicate where 'no mass enhancement' occurs.*

The suggested explanation is included in figure captions in the revised manuscript.

Referee's comment: *Figure 2: What are the shaded bands on panel d? Presumably this is not a designation of 'nighttime'. They may be present to guide the eye but are confusing and should be explained. I recommend either removing them or making them distinct from the nighttime bands in panels a-c.*

There should have been no shaded bands in panel d. We are sorry for this technical error that has been corrected in the revised manuscript.

Referee's comment: *Figure 6 and 7: It's difficult to see the dashed line for $C_{tot}/Co$ within some of the panels (e.g. Fig. 6 panel a); can this line be made more distinctive?*

Visibility of the dashed lines for $C_{tot}/C_0$ is improved in the revised manuscript by increasing their width.

Referee's comment: *Page 5 line 11: Bian et al., 2017 simulated 4 hours of aging (not 5).*

We thank the Referee for this correction. In the same sentence in the revised manuscript, we mention numerical experiments by Hodshire et al. (2019) along with those by Bian et al. (2017) and refer to the "first few hours" of BB OA evolution.

Referee's comment: *Page 8 line 17: are the SVOCs evenly distributed across the 5 bins? Please clarify.*

Actually, the SVOCs were not distributed evenly across the different volatility bins. To avoid possible confusion, the corresponding sentence has been revised. Instead of saying that the SVOCs were distributed across the 5 bins, we say that all SVOCs are represented using five volatility classes. The volatility distributions used in our experiments are specified in Sect. 2.3.

References

Bian, Q., Jathar, S. H., Kodros, J. K., Barsanti, K. C., Hatch, L. E., May, A. A., Kreidenweis, S. M., and Pierce, J. R.: Secondary organic aerosol formation in biomass-burning plumes: theoretical analysis of lab studies and ambient plumes, Atmos. Chem. Phys., 17, 5459-5475, https://doi.org/10.5194/acp-17-5459-2017, 2017.

Hodshire, A. L., Bian, Q., Ramnarine, E., Lonsdale, C. R., Alvarado, M. J., Kreidenweis, S. M., Jathar, S. H., and Pierce, J. R.: More than emissions and chemistry: Fire size, dilution, and background aerosol also greatly influence near-field biomass burning aerosol aging, J. Geophys. Res. Atmos., 124, 10, 5589-5611, https://doi.org/10.1029/2018JD029674, 2019.

Konovalov, I. B., Berezin, E. V., and Beekmann, M.: Effect of photochemical self-action of carbon-containing aerosol: wildfires, Izv. Atmos. Ocean. Phy., 52, 263–270, https://doi.org/10.1134/S0001433816030063, 2016.

Mikhailov, E. F., Mironova, S., Mironov, G., Vlasenko, S., Panov, A., Chi, X., Walter, D., Carbone, S., Artaxo, P., Heimann, M., Lavric, J., Pöschl, U., and Andreae, M. O.: Long-term measurements (2010–2014) of carbonaceous aerosol and carbon monoxide at the Zotino Tall Tower Observatory (ZOTTO) in central Siberia, Atmos. Chem. Phys., 17, 14365-14392, https://doi.org/10.5194/acp-17-14365-2017, 2017.

Paris, J.-D., Stohl, A., Nédélec, P., Arshinov, M. Yu., Panchenko, M. V., Shmargunov, V. P., Law, K. S., Belan, B. D., and Ciais, P.: Wildfire smoke in the Siberian Arctic in summer: source characterization and plume evolution from airborne measurements, Atmos. Chem. Phys., 9, 9315-9327, https://doi.org/10.5194/acp-9-9315-2009, 2009.

Wang, Q., Saturno, J., Chi, X., Walter, D., Lavric, J. V., Moran-Zuloaga, D., Ditas, F., Pöhlker, C., Brito, J., Carbone, S., Artaxo, P., and Andreae, M. O.: Modeling investigation of light-absorbing aerosols in the Amazon Basin during the wet season, Atmos. Chem. Phys., 16, 14775-14794, https://doi.org/10.5194/acp-16-14775-2016, 2016.

---

## Author Comment (AC2) · 25 Aug 2019

Referee's comment: *The use of chemistry transport models to simulate a single plume seems to be off-scale. Would the authors comment on the uncertainties associated with this disadvantage?*

We agree with the Referee that chemistry transport models (CTMs) are not designed to simulate a single BB plume and should not normally be used for this purpose. However, in practice, CTMs are generally applied to situations where the actual spatial inhomogeneity of BB OA emissions is not resolved in the simulations. This is what we meant in our remark about simulations of BB OA evolution with CTMs in the introduction ("While three-dimensional chemistry transport models are intended to provide the best possible quantitative representation of the evolution of OA and its gaseous precursors…"). To make our point clearer and to address the Referee's comment, the following sentence has been included in the Introduction of the revised manuscript: "Note that while the spatial scales representative of isolated BB plumes are typically not resolved by chemistry transport models, simulations of a single BB plume with a box model can provide useful insights into possible uncertainties introduced by neglecting the spatial inhomogeneity of BB OA emissions in chemistry transport models at the sub-grid scales." The nature of these possible uncertainties is further discussed in Sect. 4.

Referee's comment: *Page 10: Would the authors comment on why they did not include NVSOA formation in MDMOA? Or is it the conventional OA scheme that was mentioned on page 8 line 6?*

The main reason why we disregarded condensed-phase processes and NVSOA formation in this study was briefly explained in the paragraph devoted to the description of the K15 scheme ("In view of the lack of robust knowledge about the condensed-phase processes (see also Section 4) and for consistency with the other numerical experiments performed in the present study, the transformation of SOA into NVSOA has been disregarded in our simulations."). In the revised manuscript, we provide a similar explanation but concerning the S15 scheme. Our choice of the FragSVSOA configuration for our experiments is further justified in the revised manuscript (specifically, in the description of the S15 scheme). In particular, we note that the FragSVSOA configuration (where NVSOA formation is disregarded) enables better consistency of the S15 scheme with the other VBS schemes considered in our study, and thus any differences between the simulations performed with the S15 scheme and the other schemes are easier to interpret. We also note that possible formation of NVSOA due to particle-phase reactions is among the factors (discussed more in detail in Sect. 4) that can affect the real BB OA evolution, but which were not analyzed in our study, as it is focused on identification of major qualitative nonlinear effects in the BB OA behavior due to gas-phase oxidation reactions in BB plumes.

Referee's comment: *Page 13 line 24-26: Would the authors specify in the text each of the size bin's range? Also, which 3 size bins were used for T18 and T18f?*

In the revised manuscript, we specified that in the experiments with the C17, K15, S15, and LIN schemes, the aerosol size distribution included 9 size bins covering the range from 20 nm to 10 µm and following a geometric progression with the common ratio of $500^{1/9}$ (~2.0), while the experiments with the T18 and T18f schemes were conducted using only 3 size bins that were defined to cover the same range (from 20 nm to 10 µm) using a geometric progression with the common ratio of $500^{1/3}$ (~7.9). We believe that given this information, a reader can easily evaluate each of the size bin's ranges.

Referee's comment: *Page 14 lines 3-7: "The concentration of OH .... based on the ambient measurements by Akagi et al. (2012), its value was set to $5 \times 10^6$ cm$^{-3}$ in all our simulations. We also assumed a constant temperature of 298 K... " Is a plume environment equivalent to ambient conditions? e.g. is it valid to assume a constant temperature of 298K and ambient OH*

*concentration in a plume? Would limited photochemistry within a plume reduce OH concentration?*

We are sorry for a somewhat misleading word "ambient" which we used to characterize the OH concentration measurements by Akagi et al. (2012). Actually, the OH concentration was indirectly measured by Akagi et al. inside of a BB plume. And indeed, the limited photochemistry within a plume is likely to reduce OH concentration. In the revised manuscript, our assumptions concerning the OH concentration and temperature and corresponding limitations of our study are explained more clearly. In particular, we note that the OH concentration within a plume can be affected by many factors (such as, e.g., the UV flux, the concentrations of nitrogen oxides and VOCs within the plume) which can cause variability of the OH concentration level across different plumes as well as temporal and spatial fluctuations of OH concentration within a given plume. We note further that temperature is also likely to vary, both spatially and temporally, within real-world BB plumes: in particular, it is likely to be lower in the upper part of a plume than near the surface. Finally, we argue that although all possible variability and inhomogeneities of the OH concentration and temperature were disregarded in our simulations, this limitation allowed us to isolate and investigate the internal dynamics of the BB OA system under fixed pre-defined conditions.

Referee's comment: *Page 14 lines 8: "Along with aerosol species, MDMOA has been configured to simulate the evolution of an inert tracer." Would the authors please clarify the concept and use of an inert tracer (it was initially mentioned in the abstract)? What is its composition and properties?*

The requested clarification concerning the use and properties of an inert tracer is provided in the revised manuscript (Sect. 2.3). In particular, we explain that the tracer is intended to represent the evolution of the BB OA mass concentration in a hypothetical situation where BB aerosol is composed of chemically inert and non-volatile components, and so the tracer was introduced in our model as a chemically inert species which can be affected only by the dilution process (since the dry and wet deposition processes were not considered in our simulations); for definiteness, the molecular weight of the tracer has been set to be the same as of carbon monoxide (CO). We note that the concept of analyzing the evolution of BB aerosol versus the evolution of an inert tracer (usually represented by CO) has been fruitfully exploited in many previous experimental and modeling studies of BB aerosol. The corresponding references are provided in the revised manuscript.

Referee's comment: *Page 19, Figure 4, perhaps setting both figures with the same y-axis scale would be helpful. Also, in the text, it's unclear why only T18 and T18f schemes are shown, but not other schemes.*

Figure 4 is redrawn with the same y-axis scale, as suggested by the Referee. We also provided an additional plot (in the same figure) showing the simulations with the C17 scheme. At the end of Sect. 2.3 of the revised manuscript, we explain that the hygroscopicity parameter was calculated only with the C17, T18 and T18f schemes because the other oxidation schemes (K15, S17 and LIN) considered in our study are not designed to evaluate the O:C ratio.

Referee's comment: *Page 24 "This observation indicates that the mass concentration of aged BB OA is likely to be much more strongly affected in the simulations by uncertainties in available representations of the BB OA evolution than the mass concentration of relatively fresh BB OA. One of the reasons is that fragmentation reactions become increasingly important with time when the SOG oxidation level increases, and then the competition between functionalization and fragmentation creates the more complex dependence on the plume parameters." Would the*

*authors please elaborate on or quantify the competition between functionalization and fragmentation? e.g. the branching ratios?*

Indeed, the competition between functionalization and fragmentation can be quantified using the fragmentation branching ratio. To address the Referee's comment, we have modified one of the sentences cited by the Referee and included an additional sentence: "As a result of this competition, the outcome of the BB OA evolution becomes strongly dependent on the fragmentation branching ratio associated with a given OA scheme."

Referee's comment: *Page 32 lines 28-29: The authors mentioned that such differences among VBS are under the "typical conditions in summer mid-latitudes", it would certainly be intriguing for future studies to examine how these behaviors vary in different environments.*

We agree with the Referee and hope that future studies can use a similar approach to examine BB OA evolution under a wider range of environmental conditions. A corresponding remark is included in Conclusions.

Referee's comment: *Figures comments: It's unclear what the shaded greys represent in the figures, and they are quite distracting, I suggest that the authors justify them in the caption or remove them.*

The meaning of the shaded grey bands was explained in the caption for Figure 2 ("Shaded bands depict nighttime periods when no oxidation reactions were allowed to occur."). The same explanation is included in the captions for Figures 3, 4, and 10.

Referee's comment: *It may be already sufficient that figures have different colored lines. Adding different symbols are just adding noise to the figures (just as with the grey shades). But it could be a personal preference, just a suggestion.*

We appreciate the Referee's opinion, but we presume that some potential readers of our paper will prefer to print it out on a black and white printer that makes different colored lines indistinguishable. For such readers, we marked the curves with different symbols.

Referee's comment: *Again, it would be helpful to compare different schemes if the axis scales are the same when possible (in the same magnitude range).*

To address the referee's comment, we modified Figures 4, 6 and 7 (except for the panel b) accordingly. Note that not all of the panels have the same scale in other figures because our main intention was to clearly demonstrate the distinctive qualitative (rather than quantitative) features of the simulated BB OA evolution under different values of parameters.

Referee's comment: *For consistency among figures, I would suggest that Figure 4 follow other figures (Fig 3 etc) to include legends in the same locations.*

We thank the Referee for this reasonable suggestion. We have placed one of the figure legends (indicating a scheme) in Figure 4 into the left upper corner, similar to the legends in Figures 2, 3, and 5. Given the large size of the other legend in Figure 4, we preferred to show this legend in a separate panel. Most of the other figures have the legends placed uniformly into the right upper corner, as in many instances the left upper corner is filled with curves. We found that using the same layout for all the figures would not ensure the best visibility for our findings.

Referee's comment: *Figure 6's (and Figure 7's) caption mentions "dashed lines," although at first read (with-out looking at the legend's $C_{tot}/C_0$), it's unclear if it's the dashed lines with dots and dashes, or dashes alone (which should be $C_{tot}/C_0$). There two types of dashes, it'd be helpful to distinguish the two.*

We have made the dashed lines for $C_{tot}/C_0$ and dash-dot lines for "neutral" enhancements more distinguishable in the revised manuscript by increasing and decreasing the lengths of the dashes for the dash-dot and dashed lines, respectively.

Reference

Akagi, S. K., Craven, J. S., Taylor, J. W., McMeeking, G. R., Yokelson, R. J., Burling, I. R., Urbanski, S. P., Wold, C. E., Seinfeld, J. H., Coe, H., Alvarado, M. J., and Weise, D. R.: Evolution of trace gases and particles emitted by a chaparral fire in California, Atmos. Chem. Phys., 12, 1397–1421, https://doi.org/10.5194/acp-12-1397-2012, 2012.